# 🏔 SWE-Milestone: Evaluating AI Agents on Continuous Software Evolution

**Gangda Deng** [* 1] **Zhaoling Chen** [* 2] **Zhongming Yu** [3] **Haoyang Fan** [1] **Yuhong Liu** [1] **Yuxin Yang** [1] **Dhruv Parikh** [1]
**Rajgopal Kannan** [4] **Le Cong** [5] **Mengdi Wang** [6] **Qian Zhang** [2] **Viktor Prasanna** [1] **Xiangru Tang** [† 7] **Xingyao Wang** [8]

## Abstract

Real-world software must continuously evolve to meet ever-changing and open-ended requirements. AI agents, increasingly deployed as long-running systems, are now entrusted to drive this evolution. Yet, existing benchmarks evaluate agents on isolated, one-off coding tasks, neglecting the temporal dependencies and technical debt inherent in real-world software evolution. To bridge this gap, we introduce DeepCommit, an agentic pipeline that reconstructs verifiable Milestone DAGs from noisy commit logs, where milestones are defined as functionally cohesive development goals. These executable sequences enable SWE-Milestone, a benchmark that evaluates agents on streams of milestone-level tasks, requiring them to sustain system integrity and limit error accumulation, dimensions of long-term software evolution largely missing from current benchmarks. Our evaluation of 12 frontier models across 4 agent frameworks reveals a critical vulnerability: overall performance scores drop significantly from >80% on isolated tasks to 38.03% in continuous settings, exposing agents' profound struggle with long-term maintenance and error propagation. Project website: swe-milestone.com.

## 1. Introduction

Software development in the real world is driven by dynamic, open-ended requirements. New requirements continuously emerge, with some building on earlier ones while others can be pursued in parallel. As a result, software evolves through an ongoing, incremental process rather than a one-time effort. Frontier LLM agents (e.g., Claude Code (Anthropic, 2025), Codex (OpenAI, 2025)) are increasingly entrusted to drive this evolution as long-running systems (OpenClaw, 2026; Nous Research, 2026), autonomously developing and refining software within complex environments rather than producing one-off edits. Over time, such an agent's accumulated efforts naturally trace out a complete repository evolution history.

Yet, evaluation for such long-running agent systems remains largely under-explored. While benchmarks for agents on coding tasks have advanced from isolated function completion (Chen et al., 2021) to full-scale codebase generation (Zhao et al., 2025; Yang et al., 2026) (Table 1), they predominantly treat development as independent tasks (Jimenez et al., 2024; Deng et al., 2025; Zan et al., 2025). A critical dimension remains unaddressed: the temporal structure of software evolution. A true *repository evolution* benchmark must capture the full evolution itinerary—a continuous stream of dependent tasks where early implementation decisions constrain subsequent ones. Ignoring these dependencies allows agents to take expedient shortcuts that satisfy immediate tests but silently accumulate technical debt (Chen et al., 2026; Orlanski et al., 2026), undermining the long-term maintainability of the codebase, a failure mode that remains invisible to current isolated evaluations (Yao, 2025).

To capture these long-term dynamics, our benchmark replays the dependency-rich evolution of high-quality open-source repositories (Badertdinov et al., 2025; Pan et al., 2025; Jain et al., 2025; Fu et al., 2026) as a high-fidelity proxy for the continuous *repository evolution* that real-world software development demands. However, determining the appropriate **task granularity** is non-trivial (Figure 1). Intuitively, one might attempt to measure evolution at the **release-level** (Thai et al., 2025). However, this granularity is **too coarse**: release snapshots collapse the hundreds of interdependent commits between versions into a single update, flattening the fine-grained dependency structure that drives evolutionary changes. In contrast, the **commit-level** history (Jimenez et al., 2024; Chen et al., 2026) is

---

[*]Equal contribution. [†]Corresponding author.

DeepCommit Pipeline: github.com/DeepCommit-ai/DeepCommit.

[1]University of Southern California [2]University of California, Riverside [3]University of California, San Diego [4]DEVCOM Army Research Office [5]Stanford University [6]Princeton University [7]Haven [8]OpenHands. Correspondence to: Gangda Deng <gangdade@usc.edu>, Zhaoling Chen <zchen526@ucr.edu>, Xiangru Tang <xiangru.tang@yale.edu>.

*Proceedings of the 43rd International Conference on Machine Learning*, Seoul, South Korea. PMLR 306, 2026. Copyright 2026 by the author(s).

*Figure 1.* Milestone-level task granularity optimally balances functional coherence and evolutionary awareness for benchmarking continuous software evolution.

**too fine-grained and imbalanced**: many commits are trivial (e.g., typo fixes) while a few are substantive, and the linear commit sequence encodes only chronological apply order, introducing spurious dependencies between unrelated changes (Herzig et al., 2016; Fan et al., 2024).

To address this, we propose modeling software evolution at the **Milestone-level**. We define a milestone as a coherent functional unit that preserves dependency constraints. This granularity strikes a balance: unlike releases, it retains the fine-grained dependencies and structural evolution of the codebase, and unlike commits, it encapsulates realistic and coherent functional goals. Functional dependencies among milestones form a Directed Acyclic Graph (DAG), which captures prerequisite constraints while allowing independent features to proceed in parallel. However, constructing Milestone DAGs requires reordering and grouping commits, which disrupts the native git history. This poses a severe challenge to correctness: applying reordered patches often breaks **compilation** and **test collection**, jeopardizing the benchmark's executability and realism.

To resolve this, we introduce **DeepCommit**, an automated agentic pipeline that reconstructs *verifiable* software evolution itineraries in the form of *Milestone DAGs*. By synergizing static analysis, LLM-agent-driven milestone construction, and runtime validation, DeepCommit ensures the synthesized milestones are executable and testable. Powered by `Claude Opus 4.5`, it achieves a high average test collection success rate of 87.1%, providing broad verification coverage. Designed as a scalable agentic framework, DeepCommit is poised to leverage future LLM advancements to harvest increasingly accurate and extensive evolution itineraries from the vast open-source ecosystem (Pan et al., 2025; Jain et al., 2025; Fu et al., 2026).

Building on this foundation, we present **SWE-Milestone**, a benchmark that operationalizes this milestone-level granularity for evaluating LLM agents under continuous software evolution. SWE-Milestone comprises 98 human-verified milestones across 7 evolution itineraries (Milestone DAGs), each from a release range of a unique high-impact open-source repository, and spanning five programming languages. Rather than solving independent tasks, agents in SWE-Milestone are tasked with evolving a codebase through streams of these dependency-constrained milestones, closely mirroring real-world development scenarios. A single full evaluation costs approximately $500 with frontier models such as `Claude Opus 4.5`. To achieve a high score in this setting, an agent must maintain long-term context, manage architectural consistency, and prevent error accumulation across extended development horizons.

Using SWE-Milestone, we conduct an evaluation of 4 frontier agent frameworks and 12 state-of-the-art LLMs. We assess performance using a unified **Score** (Section 5.1), which balances **Recall** (completeness of new feature implementation) and **Precision** (robustness against regressions), along with a strict **Resolve Rate** for fully completed milestones. Our evaluation reveals the following key findings regarding agent capabilities in continuous software evolution:

- **A fundamental performance gap: Continuous vs. Independent.** (Section 5.2) Frontier models exhibit a substantial degradation from independent to continuous task evaluation. Scores drop from over 80% on isolated tasks to at most 38.03% (`Claude Opus 4.6`) in continuous environments, with a mere 13.37% Resolve Rate (`Gemini 3 Pro`).

- **Recall grows linearly but Precision saturates.** We identify a fundamental asymmetry in continuous software evolution (Section 5.3): while frontier agents retain the capability to implement new features (linear Recall growth), they fail to prevent regressions as the system evolves (saturated Precision). This indicates that agents struggle primarily with system-level maintenance rather than local implementation.

- **Accumulated errors stall downstream progress.** Unresolved regressions snowball faster than agents can fix them (Appendix D.3), propagating through dependency chains until downstream development stalls.

*Table 1.* Representative software engineering benchmarks for LLMs. Unlike other categories of benchmarks that evaluate agents on isolated snapshots or against ground-truth states at each step, *Repository Evolution* requires agents to continuously build upon their own accumulated development history, exposing them to error propagation across tasks. SWE-Milestone adopts *Milestone-level* granularity: each milestone is a functionally coherent group of commits that collectively advances a development objective, avoiding the noise of individual commits and the excessive scope of full releases. *Dev History* denotes the agent's own development trace accumulated from preceding tasks.

| Category | Benchmark | Language | Task Properties | | | Cross-task Dependency | Task Collection |
|---|---|---|---|---|---|---|---|
| | | | Granularity | Avg LoC | Additional Context | | |
| Function Completion | HumanEval (Chen et al., 2021) | Python | Function-level | 6.8 | – | – | Manual |
| Issue Resolution | SWE-bench (Jimenez et al., 2024) | Python | Commit-level | 32.8 | Codebase | – | Rule-based |
| | SWE-rebench (Badertdinov et al., 2025) | Python | Commit-level | 142 | Codebase | – | Rule-based |
| | Multi-SWE-bench (Zan et al., 2025) | Multi | Commit-level | 246 | Codebase | – | Rule-based |
| | SWE-bench Pro (Deng et al., 2025) | Multi | Commit-level | 107 | Codebase | – | Rule-based |
| | SWE-CI (Chen et al., 2026) | Python | Commit-level | ∼60 | Codebase + Oracle Test | Commit Chain | Rule-based |
| Codebase Generation | SWE-Dev (Du et al., 2026) | Python | Feature-level | 190 | Codebase | – | Rule-based |
| | FeatureBench (Zhou et al., 2026) | Python | Feature-level | 790 | Codebase | – | Test-driven |
| | SWE-EVO (Thai et al., 2025) | Python | Release-level | 611 | Codebase | – | Rule-based |
| | Commit0 (Zhao et al., 2025) | Python | Project-level | >3k | Codebase Skeleton | – | Rule-based |
| | NL2Repo (Ding et al., 2026) | Python | Project-level | >3k | – | – | Rule-based |
| | ProgramBench (Yang et al., 2026) | Multi | Project-level | >8k | Executable | – | Agentic |
| **Repository Evolution** | **SWE-Milestone (Ours)** | **Multi** | **Milestone-level** | **570** | **Codebase + Dev History** | **Milestone DAG** | **Agentic** |

- **Proactive exploration and verification mitigate technical debt.** Behavioral analysis shows that successful sustained evolution relies on proactive codebase exploration and disciplined test verification, whereas both blind trial-and-error and the absence of verification accelerate failure (Appendix D.4).

**Conflict of Interest Disclosure.** One author (X.W.) is affiliated with OpenHands, which is among the evaluated agent frameworks. All metrics and protocols are applied uniformly across frameworks.

## 2. Related Work

We focus here on the three SWE benchmark families most closely related to SWE-Milestone: issue resolution, codebase generation, and continuous SWE tasks. Additional context on LLM-driven coding agents, automated SWE environment synthesis, and performance-optimization benchmarks is provided in Appendix A.

**Issue Resolution Tasks**. Unlike Terminal-Bench (Merrill et al., 2026), which tests agents on shell commands isolated from any codebase, SWE benchmarks require modifying real-world GitHub repositories. SWE-bench (Jimenez et al., 2024) pioneered this line by pairing issues with hidden tests, followed by SWE-bench Pro (Deng et al., 2025) for data hygiene and Multi-SWE-bench (Zan et al., 2025) for multilingual coverage.

**Codebase Generation Tasks**. Codebase generation benchmarks pursue longer-horizon evaluation by progressively enlarging the scope of a single task. At the feature level, SWE-Dev (Du et al., 2026) and FeatureBench (Zhou et al., 2026) require agents to implement complete features against curated test suites. At the release level, SWE-EVO (Thai et al., 2025) bundles all changes between consecutive release tags into a single task. At the repository level, Commit0 (Zhao et al., 2025), NL2Repo (Ding et al., 2026), and ProgramBench (Yang et al., 2026) push agents toward synthesizing entire repositories from scratch. Despite the enlarged scope, these benchmarks evaluate task instances in isolation on a reset codebase, leaving inter-task dependencies and accumulated technical debt unmodeled. SWE-Milestone instead links milestones through a dependency DAG over a persistent repository, making both intermediate progress and cross-task error propagation directly measurable.

**Continuous SWE Tasks**. Real software development involves a stream of dependent tasks on the same codebase. While LLMs are known to degrade in this setting relative to single-shot evaluation (Laban et al., 2026), dedicated benchmarks remain nascent. SlopCodeBench (Orlanski et al., 2026) measures structural erosion and verbosity drift on hand-crafted tasks where an agent extends a codebase built from scratch. A complementary thread evaluates agents via continuous integration (CI) loops on real repositories (Xu et al., 2026). SWE-CI (Chen et al., 2026), for instance, chains commit-to-commit CI rounds by exposing ground-truth tests to a dual-agent protocol following the native commit order. SWE-Milestone instead groups commits into functionally coherent, self-contained milestones whose dependencies form a DAG, better modeling continuous evolution than scattered commit-level CI.

## 3. DeepCommit: An Automated Pipeline for Reconstructing Software Evolution

Software repositories encode rich evolutionary trajectories, yet raw commit histories remain noisy, fragmented, and inadequate as executable development sequences.

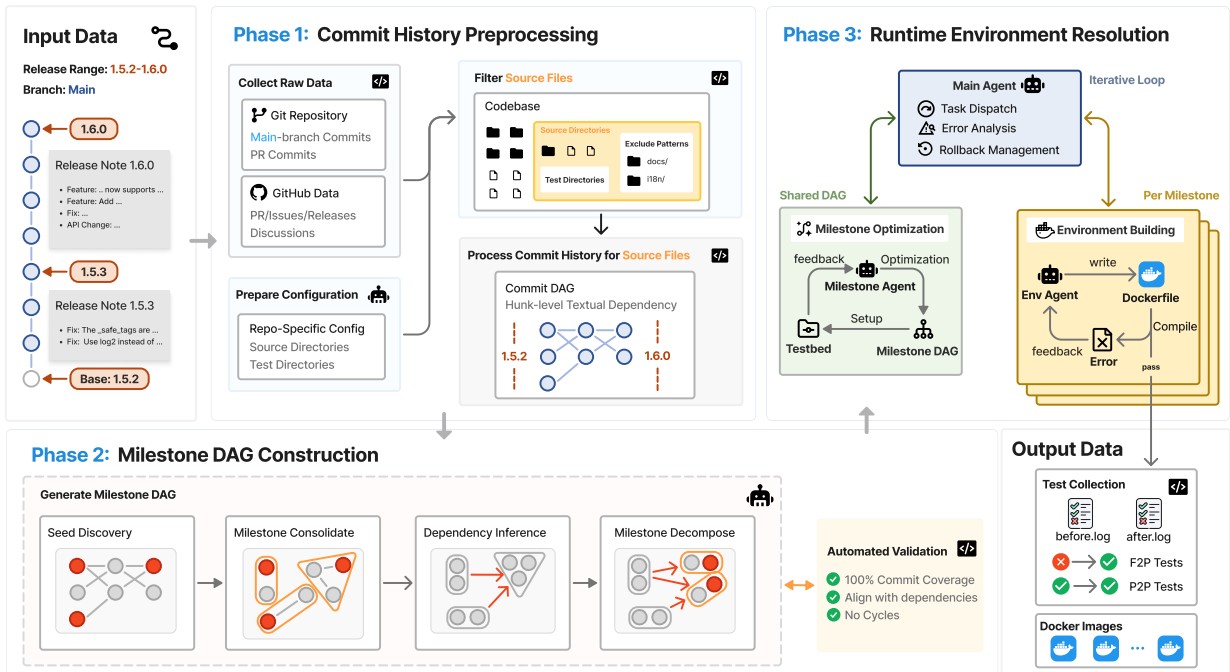

*Figure 2.* The DeepCommit pipeline architecture. **Phase 1** extracts structured data from commit history through static analysis, including source filtering, commit extraction, PR/Issues, releases, commit DAG, code metrics, and symbol changes. **Phase 2** employs an LLM agent to construct a Milestone DAG via four iterative stages: seed discovery, milestone consolidation, dependency inference, and milestone decomposition. **Phase 3** resolves runtime dependencies through testbed construction and test collection, with DAG refinement and fallback patches as repair strategies, followed by flaky test filtering to produce an executable testbed. **Quality Assurance** validates outputs at textual, compilation, and test collection levels. See Appendix C.4 for all DAG visualizations.

## 3.1. From Raw Commits to Milestone DAGs

Commits vary widely in granularity, semantic clarity, and dependency structure, while parallel branches, squash merges, and non-functional changes obscure true developmental relationships. Relying on documentation or release notes alone lacks sufficient resolution to reconstruct precise code evolution. DeepCommit addresses this challenge by transforming linear git histories into structured, verifiable *Milestone DAGs*, where each node represents a coherent, testable unit of development and edges encode dependency constraints across evolution phases.

## 3.2. Overall Agent-Driven Pipeline

As illustrated in Figure 2, DeepCommit reconstructs software evolution itineraries through an end-to-end pipeline that sequentially integrates: (1) commit history preprocessing, (2) Milestone DAG construction, and (3) executable environment resolution.

### 3.2.1. COMMIT HISTORY PREPROCESSING

We model each repository's main-branch range between release tags as a linear sequence of commits. This linearization aligns naturally with the squash-and-merge workflow, the most widely adopted convention in actively maintained, high-quality repositories, where each main-branch commit

maps to a Pull Request (PR) or Issue together with its internal sub-commits. We collect all main-branch commits and their PR, Issue, and Release metadata. An LLM agent then prepares a per-repository configuration of source directories, test patterns, and exclusion rules that separates product-logic source from test code and filters out commits touching only non-source files such as documentation, CI configurations, and build assets (Appendix B.1). To enable downstream milestone discovery and dependency inference, we extract three structural signals via static analysis: (i) a commit-level DAG built with `git blame` to capture line-level textual dependencies, (ii) symbol-level modifications identifying changes in classes and functions, and (iii) file-level co-change statistics reflecting evolutionary coupling.

### 3.2.2. MILESTONE DAG CONSTRUCTION

Organizing hundreds of discrete commits into functionally coherent milestones requires integrating structural dependencies with code-level reasoning. We employ a four-stage LLM-agent-driven process to progressively construct the Milestone DAG. Each stage is orchestrated with automated data preparation, agent-accessible validation tools, and a post-stage quality gate. A stage runs in a forward pass and may iterate internally against its self-checks. At stage boundaries, the pipeline can also fan out into multiple parallel instantiations of the next stage and retain the variant that

best satisfies the downstream quality gate.

**Seed Discovery.** Each milestone is initiated by a *seed commit*, namely a foundational anchor that introduces a distinct development theme. An LLM agent identifies such seeds by jointly evaluating commit semantics (commit messages and linked discussion context) and structural signals (DAG topology, including out-degree and descendant count), filtering out cosmetic edits, hotfixes, and follow-up patches that lack downstream structural influence.

**Milestone Consolidation.** For each seed, parallel sub-agents expand the milestone boundary using shared file modifications, temporal proximity, and PR/Issue references, growing each seed into a milestone that bundles all commits realizing its development theme. Since the sub-agents operate independently, the same commit may be claimed by several milestones. A coordinating agent then resolves these overlapping claims so that each eligible commit belongs to exactly one milestone, enforces complete coverage of the preprocessed commit range, and certifies acyclicity.

**Dependency Inference.** The majority of inter-milestone edges follow directly from the line-level textual dependencies extracted during preprocessing. More subtle dependencies, such as call relationships that share no common hunk, are proposed and validated by an LLM agent using symbol-level analysis and file co-change patterns. Even so, certain dependencies surface only when milestones are built and executed. These residual edges are recovered later during runtime environment resolution (Section 3.2.3).

**Milestone Decomposition.** Oversized milestones are decomposed into functionally independent sub-milestones while underspecified ones are merged into adjacent neighbors, with dependencies synchronously updated to preserve a valid DAG. When an oversized milestone is dominated by a single squashed PR-commit, the agent further re-segments the commit's diff along feature boundaries and remaps the affected dependency edges via line-level blame, promoting the resulting sub-units to first-class milestones. The pipeline targets a coefficient of variation $CV < 1.0$ over per-milestone LoC, achieving $CV = 0.96$ on SWE-Milestone (Appendix B.2).

### 3.2.3. RUNTIME ENVIRONMENT RESOLUTION

To transform the Milestone DAG into an executable evaluation environment, a *MainAgent* orchestrates a multi-agent workflow that produces, for every milestone, a reproducible Docker image that yields stable test signals. From a whole-evolution perspective, the *MainAgent* balances testbed quality against the cost of automated resolution. To achieve high quality at reasonable cost, the pipeline additionally relies on human-expert guidance to steer the *MainAgent* at key decision points. The *MainAgent* then dispatches sub-agents for batch analysis and problem localization, routing the surfaced issues to two specialized repair modules that iterate together until every milestone reaches a stable, fully collected state.

**Milestone DAG Optimization and Testbed Preparation.** A *MilestoneAgent* reconstructs each milestone's code state by cherry-picking its commits in topological order onto the codebase at the base release tag. When a cherry-pick conflict arises, the agent repairs the DAG, primarily by adding the missing cross-milestone dependency edge and, when necessary, relocating misattributed commits to the correct milestone. Commits that still cannot be applied are marked as deferred, so that every milestone is reconstructed into a complete, DAG-consistent state.

**Environment Configuration and Test Collection.** For each milestone, an *EnvAgent* generates a Dockerfile from the repository's CI/CD workflows and enforces three hard gates: the source compiles, the test framework collects successfully, and as many tests referenced by the milestone's patch as possible are captured in the collected set. A characteristic failure mode arises when the configured test environment runs ahead of the cherry-picked source state, so that functional modules referenced by the collected tests do not yet exist. These cases cannot be fixed locally and are deferred to the DAG optimization module for refinement (Appendix B.3).

### 3.3. Automated Quality Assurance

To ensure the reliability and reproducibility of our evaluation, we rigorously validate each milestone testbed and report the aggregate evidence across three core dimensions, complementing the per-milestone gates enforced in Section 3.2.3:

**Milestone Graph Validity.** We verify the structural integrity of the reconstructed history. This includes confirming *commit completeness* (100% coverage of the preprocessed, eligible commit set), *dependency consistency* (ensuring milestone dependencies respect underlying commit dependencies), and *DAG correctness* (validating acyclicity).

**Runtime Executability.** We ensure that errors stem from agent code, not infrastructure. We verify *testbed compilability* by ensuring successful build and test collection in both states. We also strictly monitor execution logs to ensure environment-induced errors remain negligible ($\leq 0.10\%$).

**Evaluation Reliability.** We assess the stability of the test suites. We achieve a high *test collection rate* (87.1%). We ensure *test consistency* by validating a negligible Pass-to-Fail rate ($\leq 0.026\%$) and filtering flaky tests through three repeated runs. Finally, we require each retained milestone to expose at least one Fail-to-Pass (F2P) or None-to-Pass (N2P) test signal (details in Appendix B.4).

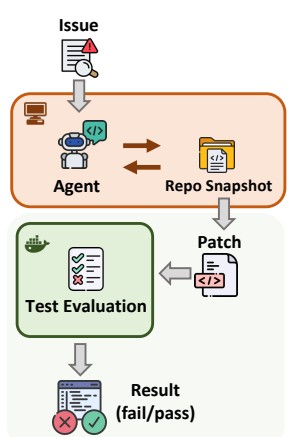
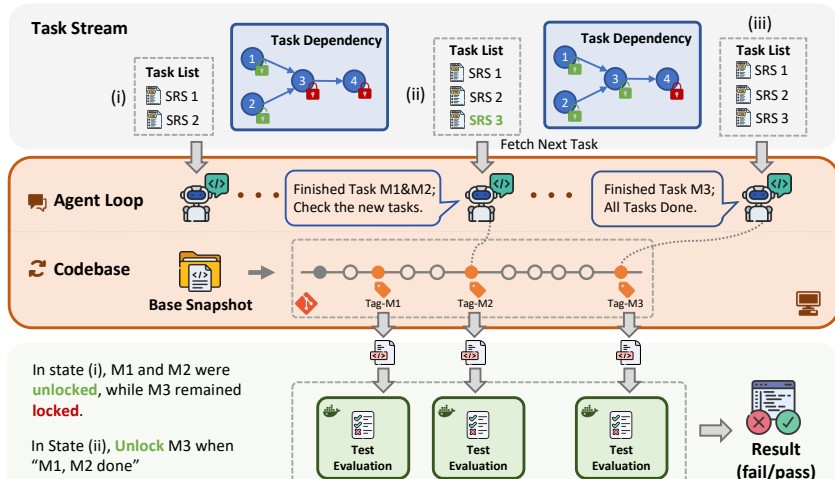

*Figure 3.* Illustration of the evaluation pipelines. (a) In the independent task evaluation workflow, the environment resets after each task. (b) In the continuous task evaluation workflow, tasks are organized as a dependency graph. The agent continuously evolves the codebase from a base snapshot. Upon completing a task (e.g., M1 & M2), the repository is snapshotted for isolated evaluation while the planner unlocks subsequent tasks (e.g., M3) for the agent to fetch, ensuring a continuous and stateful development loop.

# 4. SWE-Milestone: Benchmarking Continuous Software Evolution

SWE-Milestone introduces a novel evaluation paradigm designed to assess an agent's ability to evolve and maintain a software codebase over an extended lifecycle. As shown in Figure 3, unlike traditional benchmarks that focus on resolving independent issues, SWE-Milestone simulates a realistic, continuous development process in which requirements arrive as a stream and tasks follow DAG-structured dependency constraints, allowing independent milestones to proceed in parallel.

## 4.1. The Continuous Task Evaluation Framework

The framework orchestrates a continuous development pipeline: an external planner dynamically unlocks tasks based on a dependency graph, the agent implements them in a persistent codebase, and the framework asynchronously evaluates snapshots upon submission. This design explicitly decouples **roadmap planning** from **implementation**, allowing us to assess the agent's ability to maintain and evolve software within a structured workflow. The framework comprises three core components:

**Dependency-Driven Task Stream.** Requirements are not presented in a static batch but are unlocked dynamically. The system maintains a DAG-based task scheduler where a new milestone $M_i$ becomes available to the agent if and only if all its prerequisite milestones $\{M_j \mid M_i \ depends \ on \ M_j\}$ have been completed. This simulates real-world constraints in which foundational features must be established before dependent features are implemented.

**Continuous Evolution Environment.** The agent operates within a persistent, stateful environment where modifications from each task persist into the next. This compels the agent to maintain the long-term health of the codebase, as early technical debt or latent bugs can accumulate and impede future progress.

**Snapshot-Based Isolated Evaluation.** To combine continuous development with rigorous verification, we use a "develop-in-place, evaluate-in-isolation" strategy. After each task, the implementation is *snapshotted* and transferred to an isolated container for testing. This keeps scoring reproducible and independent of ongoing development without interrupting the agent's environment.

## 4.2. Benchmark Construction

We construct a high-quality dataset through a rigorous pipeline that transforms open-source repositories into verified evolutionary suites.

**Repository and Range Selection**. We identify projects with high community impact and diverse programming languages. We specifically select release ranges that exhibit rich dependency structures, ensuring the benchmark captures complex, non-linear development scenarios rather than trivial sequences.

**Itinerary Extraction via DeepCommit.** Leveraging the DeepCommit pipeline (Section 3), we mine the evolutionary history of selected projects. To guarantee the benchmark's quality and evaluation efficiency, we apply strict post-processing filters to the generated milestones. We retain only milestones that (1) represent core functional

changes (excluding pure documentation updates), (2) possess executable Fail-to-Pass tests as definitive success criteria, and (3) fit within a manageable context window. This step ensures that every task in the benchmark is grounded in a verified, executable state transition.

**Reverse-Engineering Software Requirements Specifications (SRSs).** Relying solely on original GitHub issues or PR descriptions is often insufficient, as they can be underspecified, outdated, or disconnected from the final code implementation. To bridge this gap, we employ an agent-driven *reverse-engineering approach* to synthesize high-fidelity software requirements specifications (SRSs). We first dispatch an LLM agent to analyze the ground-truth patches to draft precise functional requirements. This draft then undergoes a refinement phase to align acceptance criteria strictly with the verified Fail-to-Pass tests. Finally, environment-specific instructions (e.g., dependency updates) are appended by analyzing build configuration changes, ensuring a complete execution context.

**Human-in-the-Loop Verification.** Automated generation can yield logical inconsistencies and misalignment with edge cases. To mitigate this, expert annotators conduct a final review focused on *task solvability*. Annotators verify that the SRS provides all necessary information to solve the problem without leaking implementation details and that the acceptance criteria are unambiguous. Simultaneously, we validate the stability of the test suites to rule out flaky tests. This hybrid verification ensures that SWE-Milestone provides a fair assessment, distinguishing genuine agent errors from artifacts of ambiguous specifications (Appendix C.1).

### 4.3. Benchmark Statistics

SWE-Milestone comprises **98 verified milestones** across **7 diverse open-source repositories**, spanning five programming languages (Go, Rust, Java, TypeScript, Python) with a total of 109 inter-milestone dependencies. As shown in Figure 4a, the milestones are distributed across repositories with varying complexity, ranging from 9 to 23 milestones per repository. The dataset captures diverse real-world development patterns, including *major architectural changes* (e.g., multi-library support), *feature-rich iterations* (e.g., cloud-native enhancements), *stability-focused releases* (e.g., compatibility fixes), and *large-scale refactoring* (e.g., type system overhauls). This ensures SWE-Milestone evaluates agents across a broad spectrum of software engineering tasks.

Figure 4b illustrates the distribution of task complexity. The dataset exhibits substantial diversity in both specification length (SRS mean: 1,348 words) and implementation scope (gold patch LoC ranging from $< 100$ to $> 1,500$). On average, each milestone modifies 27.4 files and involves 17.1 Fail-to-Pass tests for verification alongside 6,218 Pass-to-Pass tests for regression prevention. Detailed per-repository

statistics are provided in Appendix C.3, and the full Milestone DAG visualizations for all repositories are shown in Appendix C.4.

## 5. Results and Analysis

### 5.1. Experimental Setup

**Evaluation Settings**   To isolate the effect of error accumulation, we evaluate under two settings based on the Milestone DAG: (1) **Continuous Task Evaluation**, the standard SWE-Milestone setting where agents continuously evolve a codebase under streaming requirements, and (2) **Independent Task Evaluation**, a stateless baseline (similar to SWE-bench (Jimenez et al., 2024)) in which each milestone is solved from the canonical codebase snapshot. We evaluate 12 frontier LLMs across 4 agent frameworks (Claude Code, Codex CLI, Gemini CLI, and OpenHands (Wang et al., 2025)), with full framework versions, context-management configurations, and the unified system prompt detailed in Appendix C.2.

**Evaluation Metrics**   Evaluating agents in a continuous evolution setting requires metrics that capture two competing objectives: implementing new functionality and preserving existing behavior. Traditional benchmarks such as SWE-bench rely on binary success criteria (all tests pass or fail), which are too coarse-grained to capture the nuance of incremental progress and regression. Simple pass-rate metrics conflate these two objectives, failing to distinguish an agent that implements features but introduces regressions from one that avoids regressions but makes no progress.

To address this limitation, we decompose agent performance along two complementary dimensions. **(1) Recall** measures *feature implementation completeness*, the proportion of required functional changes successfully implemented:

$$\text{Recall}_m = \frac{N_{\text{fixed},m}}{N_{\text{required},m}}, \qquad (1)$$

where $N_{\text{required},m}$ is the total number of *Fail-to-Pass* (F2P) tests for milestone $m$ (tests that transition from failing at the start state to passing after the gold patch is applied), and $N_{\text{fixed},m}$ is the count that the agent successfully fixes. **(2) Precision** measures *modification reliability*, the proportion of test status changes that are improvements rather than regressions:

$$\text{Precision}_m = \frac{N_{\text{fixed},m} + \epsilon}{N_{\text{fixed},m} + N_{\text{broken},m} + \epsilon}, \qquad (2)$$

where $N_{\text{broken},m}$ is the number of *Pass-to-Pass* (P2P) tests (passing at the start state and required to remain passing) that regress due to the agent's changes, and $\epsilon = 1$ smooths the case where the agent makes no impact. The per-milestone

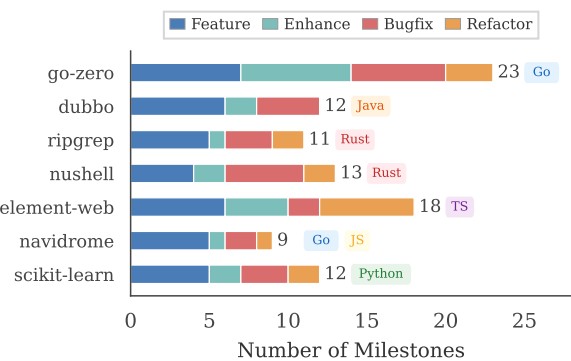

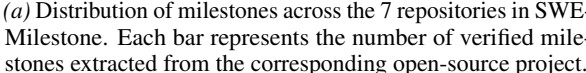

*(a)* Distribution of milestones across the 7 repositories in SWE-Milestone. Each bar represents the number of verified milestones extracted from the corresponding open-source project.

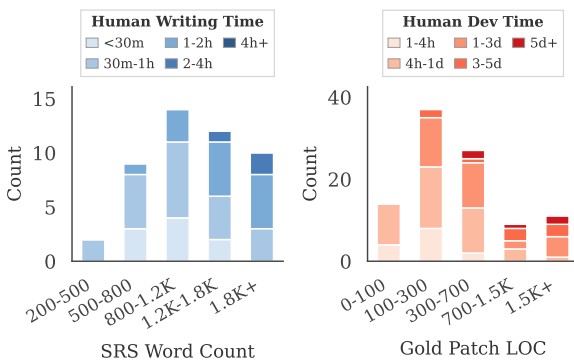

*(b)* Distribution of SRS word counts (left) and gold patch LoC (right), stratified by estimated human effort.

*Figure 4.* Dataset statistics and characteristics of SWE-Milestone.

Score is their harmonic mean (F1):

$$\text{Score}_m = \frac{2 \cdot \text{Precision}_m \cdot \text{Recall}_m}{\text{Precision}_m + \text{Recall}_m}, \quad (3)$$

For each evolution range $r$ with milestone set $M_r$, we first average milestone scores within the range and then macro-average across the set of evolution ranges $R$:

$$\text{Score}_r = \frac{1}{|M_r|} \sum_{m \in M_r} \text{Score}_m,$$
$$\text{Score} = \frac{1}{|R|} \sum_{r \in R} \text{Score}_r. \quad (4)$$

This ensures that neither dimension can be neglected: an agent that implements all features but introduces severe regressions scores as poorly as one that preserves existing functionality but makes no progress.

Consistent with prior work like SWE-bench, we also report the **Milestone Resolve Rate**, where a milestone is considered *resolved* only if the agent passes all associated F2P and P2P tests. We report the macro-average Resolve Rate across all evolution ranges. While Score quantifies partial progress, Resolve Rate assesses whether the task was fully resolved.

### 5.2. Overall Performance

Table 2 presents results across 15 agent-model configurations (12 models, 3 run under two frameworks). `Claude Opus 4.6` achieves the highest Score (38.03% in Open-Hands and 36.29% in Claude Code), followed by `Claude Sonnet 4.6` (29.58%) and `GPT 5.3-Codex` (28.88%). Across all models, the gap between Score (∼38% at best) and Resolve Rate (∼13%) is substantial: agents achieve partial progress on most milestones but rarely complete them fully. Moreover, the resolved milestones are predominantly early ones with few upstream dependencies, confirming that accumulated upstream errors increasingly hinder downstream task completion (Figure 14). Crucially, these same

milestones are largely solvable in isolation: under the **Independent Task Evaluation** setting, frontier models average >80% Score (e.g., `Claude Sonnet 4.6` reaches 93.2% on `scikit-learn`), far above the ∼38% observed here. This continuous-vs-independent gap (per-repository comparison in Figure 13, Appendix D.1) indicates the difficulty stems from long-horizon error accumulation rather than inherent task complexity. Overall, these results highlight that SWE-Milestone poses a significant challenge to current frontier models, and reliable long-horizon continuous development remains an open problem. Appendix D.1 reports generational and cost–score comparisons.

### 5.3. Evolution Dynamics: Recall Scales while Precision Saturates

Does agent capability degrade over time, or does accumulated technical debt overwhelm otherwise competent agents? We model cumulative score trajectories with a saturation function $y = a(1 - e^{-bx})$, where a small $b$ yields near-linear growth and a large $b$ saturates toward the ceiling $a$. As shown in Figure 5 (left), all continuous-evaluation models exhibit clear ceilings that multi-window extrapolation (fitting the model to progressively larger milestone subsets and projecting forward) confirms persist beyond the observed window. Comparing the two settings (middle, right; full per-repository breakdown in Figure 13, Appendix D.1), independent scores grow near-linearly while continuous scores saturate, with the gap widening monotonically. The fits separate initial efficiency from durability: `Gemini 3.1 Pro` starts fastest (init = 1.03) but retains only 1% per window, whereas `GPT 5.3-Codex` starts lower (0.68) but retains 16%. `Claude Opus 4.6` combines strong initial efficiency (0.84) with 15% retention, producing the highest projected ceiling (0.44). Thus, strong initial capability does not guarantee sustained evolution.

| Agent | Model | Score* (%)↑ | Precision (%)↑ | Recall (%)↑ | Resolve (%)↑ | Cost ($) | Out Tok. (K) | Time (h) | Turns |
|---|---|---|---|---|---|---|---|---|---|
| Claude Code | Claude Sonnet 4.5 | 15.16 | 18.88 | 28.50 | 5.49 | 27.02 | 243 | **2.06** | 770 |
| | Claude Opus 4.5 | 25.85 | 28.04 | 40.80 | 6.28 | 71.10 | 309 | 2.35 | 999 |
| | Claude Sonnet 4.6 | 29.58 | 29.62 | 47.63 | 5.88 | 68.88 | 852 | 4.41 | 1538 |
| | Claude Opus 4.6 | 36.29 | **37.84** | **56.32** | 11.57 | 88.22 | 578 | 2.73 | 1891 |
| Codex CLI | GPT 5.2-Codex | 13.46 | 12.65 | 26.65 | 4.78 | 38.11 | 701 | 3.56 | 1259 |
| | GPT 5.2 | 23.30 | 20.89 | 45.76 | 8.18 | 56.90 | 814 | 5.35 | 1717 |
| | GPT 5.3-Codex | 28.88 | 27.81 | 49.70 | 9.58 | 25.01 | 392 | 8.23 | 1109 |
| Gemini CLI | Gemini 3 Pro | 24.25 | 25.46 | 32.70 | **13.37** | 114.96 | 294 | 3.65 | **676** |
| | Gemini 3.1 Pro | 23.32 | 21.59 | 37.22 | 10.95 | 62.97 | **207** | 3.89 | 1208 |
| | Gemini 3 Flash | 24.22 | 24.31 | 42.12 | 8.37 | 12.10 | 255 | 5.02 | 1512 |
| OpenHands | MiniMax M2.5 | 17.60 | 22.48 | 34.60 | 1.30 | **3.57** | 598 | 11.88 | 1846 |
| | Kimi K2.5 | 20.20 | 26.29 | 31.37 | 8.49 | 4.32 | 279 | 6.85 | 800 |
| | Gemini 3 Flash | 22.32 | 25.20 | 37.13 | 6.59 | 16.90 | 1516 | 7.16 | 2632 |
| | GPT 5.3-Codex | 26.47 | 25.01 | 37.32 | 12.50 | 30.13 | 553 | 18.75 | 1047 |
| | Claude Opus 4.6 | **38.03** | 37.33 | 55.21 | 8.46 | 75.73 | 524 | 7.54 | 1970 |

*Table 2.* Performance on SWE-Milestone under continuous task evaluation. Metrics are per-evolution-range averages. Out Tok. (K): total generated tokens (thousands), including reasoning/thinking tokens when available. * Primary metric (shaded); **bold**: column best.

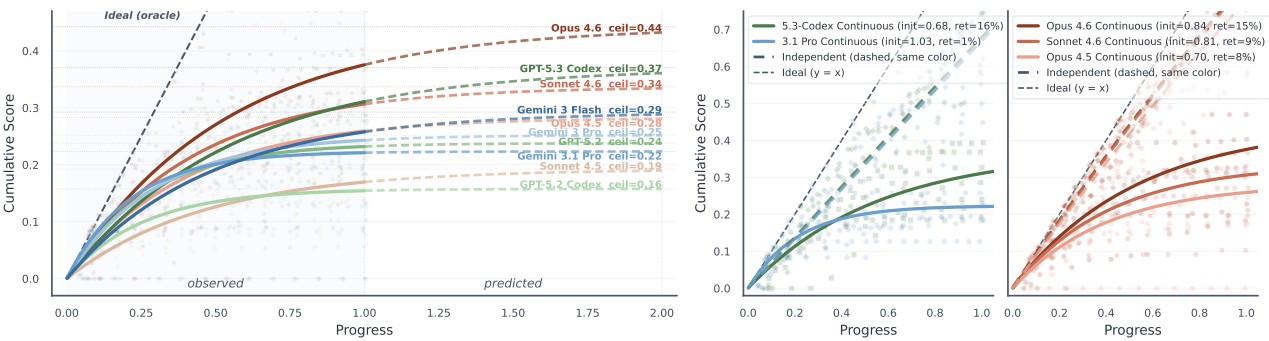

*Figure 5.* Evolution dynamics. (Left) Multi-window saturation fits $y = a(1 - e^{-bx})$ with projected ceilings; annotations give initial efficiency $\text{init} = ab$ and retained fraction $\text{retain} = e^{-b}$. (Middle) Continuous vs. Independent results for `GPT 5.3-Codex` and `Gemini 3.1 Pro`; (right) Claude models.

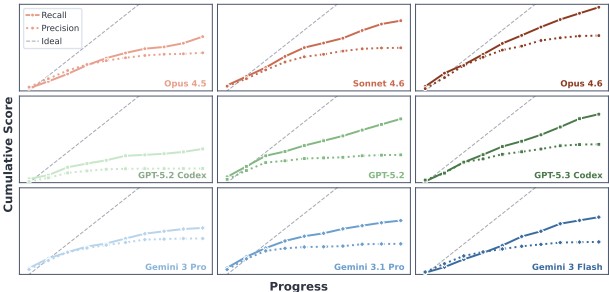

*Figure 6.* Cumulative Recall (solid) and Precision (dotted): Recall grows near-linearly, while Precision saturates.

Decomposing the cumulative score into *Recall* (feature implementation) and *Precision* (functionality preservation) isolates the mechanism (Figure 6): Recall continues to grow near-linearly across all models, while Precision saturates rapidly. The ceiling thus arises not from agents forgetting how to code, but from their inability to prevent accumulating regressions. Even stronger models saturate: unresolved regressions snowball until they overwhelm productive development.

## 5.4. Additional Analyses

Supporting analyses identify when and why agents fail. Larger patches and deeper DAGs lower Score, while upstream errors reduce later Resolve Rate (Appendix D.2). Logic Errors dominate (∼57%) and propagate downstream (Appendix D.3). Proactive exploration and moderate verification outperform blind thrashing (Appendix D.4). Deep-Commit matches human boundaries when structure is clear and otherwise follows dependency topology (Appendix D.7).

## 6. Conclusion

DeepCommit distills verifiable Milestone DAGs from noisy Git histories, enabling SWE-Milestone to benchmark continuous development. Frontier models fall from over 80% on isolated tasks to 38.03% in continuous evaluation, resolving only ∼13%. Accumulating regressions and technical debt leave sustained repository evolution an open challenge. Progress requires agents to preserve system invariants, diagnose accumulated failures, and recover from earlier errors.

## Acknowledgements

The authors of this work are supported in part by the U.S. Army Research Office under Grant W911NF-242-0194, the National Science Foundation under Grants OAC-2505107 and CCF-2426161, a Google Research Credit Award, and UCR Senate Awards. We thank OpenHands for providing the API credits that support most of the evaluations in this study. Any opinions, findings, and conclusions or recommendations expressed in this material are those of the authors and do not necessarily reflect the views of these organizations. Distribution Statement A: Approved for public release. Distribution is unlimited.

## Impact Statement

This paper presents a benchmark for evaluating LLM agents in continuous software evolution, a prerequisite for deploying autonomous agents in real-world production environments. Beyond code generation, we emphasize the ability to sustainably evolve systems over time, which is essential for long-running agent runtimes to iteratively customize software for diverse user needs. This capability unlocks a new productivity paradigm: agents serving as adaptive interfaces that bridge human intent and complex digital systems. Although highly capable coding agents may impact the software development workforce, we expect them to lower the barrier to entry for software creation, empowering a broader range of users to leverage the power of code for problem-solving.

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

# Table of Contents

## A. Additional Related Work

This appendix expands the related work in Section 2 with three threads complementary to repository-evolution benchmarks: LLM-driven coding agents, automated SWE environment synthesis, and performance-optimization benchmarks.

**LLM-Driven Coding Agents**. While basic Bash tools alone can drive an LLM through software tasks (Yang et al., 2025), specialized scaffolding is what unlocks reliable, efficient, and user-friendly behavior. Scaffolds have evolved from predefined pipelines (Xia et al., 2025) to fully autonomous systems (Yang et al., 2024; Wang et al., 2025). Commercial deployments span complementary use cases: Devin (Cognition Labs, 2024), GitHub Copilot (GitHub, 2025), Cursor (Cursor, 2024), Trae (TRAE, 2025), and Antigravity (Google, 2025a) integrate agents into IDE and cloud workflows, while Claude Code (Anthropic, 2025), Codex (OpenAI, 2025), and Gemini CLI (Google, 2025b) run in the terminal. While these agents are well-validated on independent, interactive tasks, their reliability under long-horizon, fully autonomous operation remains a formidable challenge. SWE-Milestone addresses this gap with a standardized, quantitative evaluation that surfaces fine-grained progress-level feedback.

**Automated SWE Environment Synthesis**. A growing line of work automatically constructs testable Dockerized environments from open-source repository snapshots. These environments primarily serve to continuously refresh evaluation data (Badertdinov et al., 2025; Zhang et al., 2025) and scale agent training (Pan et al., 2025; Jain et al., 2025; Fu et al.,

2026). Unlike these snapshot-based approaches, DeepCommit preserves the temporal structure of development trajectories by reorganizing commit histories into Milestone DAGs, thereby synthesizing long-horizon tasks with verifiable progress checkpoints.

**Performance Optimization Tasks**. A parallel direction evaluates whether agents can speed up code across repositories (He et al., 2025; Ma et al., 2025; Shetty et al., 2025), numerical algorithms (Press et al., 2025), and GPU kernels (Ouyang et al., 2025). While these works instantiate long-horizon evaluation, they focus on closed-ended optimization objectives, such as wall-clock time against an expert reference. In contrast, SWE-Milestone addresses the open-ended challenge of general functional software development.

## B. DeepCommit Pipeline Implementation Details

This appendix documents the per-stage implementation of the DeepCommit pipeline introduced in Section 3, covering commit history preprocessing, Milestone DAG construction, runtime environment resolution, and testbed validation.

### B.1. Commit History Preprocessing Details

In open-source projects, version tags are applied in various ways, posing challenges for determining the mainline commit range. Commonly, tags are placed directly on the main branch, making the commits between start and end tags the target range. However, many projects use a release branch model: a release branch is created from main for stabilization before release, and the tag is ultimately placed on the release branch rather than main. In such cases, directly comparing two tags would include release branch commits while missing parallel development on main.

To address this, we employ a *branch-out/first-parent* strategy to recover the mainline range. First, we use `git merge-base` to find each tag's branch-out point from the main branch. Then, we collect all commits between these two branch-out points along the main branch's first-parent chain. First-parent traversal ensures we only follow direct commits on main, ignoring internal commits from merged feature branches, thus obtaining a clean mainline evolution sequence.

**Data Collection.** For each entity, we collect metadata and content. For commits, we record change details and parent relationships. For PRs, we record descriptions, constituent commits, linked Issues, and review discussions. For Issues, we record descriptions and discussions. For Releases, we record version tags and notes.

**Per-Repo Configuration.** Source-file filtering is driven by a small per-repository configuration with four fields: `repo_src_dirs` (source-module directories to retain), `test_dirs` (test-file glob patterns), `exclude` (build artifacts and generated files to strip from within source directories), and `main_branch` (the branch used for range extraction). An LLM agent bootstraps this configuration by detecting the repository language from build markers (e.g., `pyproject.toml`, `pom.xml`, `go.mod`) and proposing source and test patterns from the repository layout and tag-range history. A human curator then verifies coverage, confirming which directories constitute product-logic source code. The rules below apply on top of this configuration.

**Filtering Rules.** Raw data contains numerous changes irrelevant to core functionality. We apply multi-layer filtering:

1. **Source directory whitelist**: Retain only changes under designated source directories (e.g., `src/`, `lib/`), excluding documentation, configuration files, and CI/CD scripts.

2. **Test file exclusion**: Exclude test code via filename patterns (e.g., `*_test.go`, `test_*.py`) and directory patterns (e.g., `tests/`, `__tests__/`).

3. **Empty commit removal**: Commits containing no valid source changes after filtering are removed.

4. **Reference consistency**: When commits are removed, the system checks PR and Issue references, removing orphaned PRs and Issues no longer referenced by any commit.

### B.2. Milestone DAG Construction Details

**Topological Features for Seed Discovery.** We utilize the following Commit DAG topological metrics to identify milestone seeds: (1) *Out-degree*: commits depended upon by multiple subsequent commits typically introduce foundational

functionality. (2) *Topological level*: commits at higher levels represent later architectural changes. (3) *Descendant count*: commits with many descendants have broad impact and are often key nodes. The agent typically identifies fewer than 20 seed commits.

**Expansion Rules for Milestone Consolidation.**    Sub-agents expand seed boundaries using the following rules: (1) *File modification overlap*: commits modifying the same files likely belong to the same feature. (2) *Temporal proximity*: commits close in time are more likely part of the same development task. (3) *Semantic association*: commit messages containing similar keywords or referencing the same Issue/PR. The system first pre-groups seeds based on commit subgraph overlap.

**Heuristics for Dependency Inference.**    For potential dependencies not covered by structural analysis, the system generates and ranks candidate edges using heuristic rules: (1) *File overlap ratio*: the proportion of shared files modified by two milestones. (2) *Symbol references*: whether downstream milestones call functions/classes defined in upstream ones. (3) *Temporal ordering*: whether the upstream milestone's median commit time precedes the downstream's. (4) *Author overlap*: milestones by the same author are more likely to have dependencies. After agent verification, dependencies are classified into two types. *Strong dependencies* indicate that missing upstream milestones will cause downstream build or test failures. *Weak dependencies* indicate functionality can degrade gracefully without affecting basic operation.

**Balancing Strategy for Milestone Decomposition.**    The system computes the source lines of code (LoC) for each milestone and uses the coefficient of variation (CV = std/mean) to measure granularity uniformity, targeting CV < 1.0. On SWE-Milestone, the final partition achieves CV = 0.96. Abnormally sized milestones are flagged in two cases: *too large*, when LoC exceeds mean $+\sigma$, and *too small*, when LoC falls below mean $-\sigma$ and under 100 lines.

**PR-Commit Splitting.**    An oversized milestone is sometimes dominated by a single squashed PR-commit, which cannot be rebalanced by regrouping commits. In this case, an LLM agent re-segments the squashed diff into smaller commits along feature boundaries, optionally emitting one integration-test commit. Dependencies among the resulting pieces, together with edges to downstream milestones, are recomputed via line-level `git blame`, so each split unit inherits only the edges it truly depends on, and the units are promoted to first-class milestones in the DAG. When no clear feature boundary exists, the agent falls back to chronological grouping, and test-only commits are never emitted as standalone milestones.

## B.3. Runtime Environment Resolution Details

**Motivation for Testbed Reconstruction.**    The original repository's commit history reflects the actual chronological order of development, not necessarily the logical dependencies between functional modules. For example, a change for dependent feature B may be committed before the prerequisite change for feature A is merged, even though B logically depends on A. Using the original history directly would cause milestone code states to be inconsistent with the DAG's dependency structure. Therefore, we re-cherry-pick commits according to the DAG's topological order, generating a new history organized by functional logic.

**Flaky Test Filtering.**    Flaky tests produce inconsistent results on the same code, severely affecting evaluation reliability. We identify such tests by running the test suite three times. If a test produces different results across runs (e.g., sometimes passing, sometimes failing), it is marked as flaky and excluded. Common causes of flakiness include: dependencies on external services, time-sensitive assertions, race conditions in concurrent code, and random data generation.

## B.4. Testbed Validation

To ensure the reliability and reproducibility of our evaluation, we rigorously validate each milestone testbed across three core dimensions: milestone graph validity, runtime executability, and evaluation reliability.

### B.4.1. MILESTONE GRAPH VALIDITY

We first verify the structural properties of the reconstructed history by construction. **Commit Completeness:** The commit set across all milestones is cross-checked against the preprocessed eligible commits from `git log v_start..v_end`, confirming 100% coverage of that filtered set with no gaps. **Dependency Consistency:** If commit $a$ in milestone $A$ depends on commit $b$ in milestone $B$ according to the commit-level DAG, then $A$ must depend on $B$ in the Milestone DAG. This constraint is verified for all inter-milestone commit pairs. **DAG Correctness:** DFS cycle detection and Kahn's topological

sort confirm valid acyclic DAGs for all 7 repositories.

### B.4.2. RUNTIME EXECUTABILITY

A reliable testbed must guarantee that errors stem from agent code, not infrastructure. Following the Milestone DAG, we reconstruct repository states by sequentially cherry-picking milestone commits in topological order. **Testbed Compilability:** We verify that `docker build` completes without error and that the test framework's collection command (e.g., `pytest --collect-only`) succeeds in both START and END states for all repositories in SWE-Milestone. **No Environment-Induced Errors:** We strictly monitor execution logs to ensure tests classified as `error` (reflecting environment/setup failures rather than assertion failures) remain negligible ($\leq 0.10\%$ across all repositories).

### B.4.3. EVALUATION RELIABILITY

Finally, we assess the runtime behavior of the test suites to ensure they provide accurate evaluation signals. **Test Collection and Size Stability:** We statically extract test names from commit diffs and match them against runtime-collected node IDs, achieving an overall collection rate of 87.1% (3,563 of 4,090 tests). Furthermore, we compare test counts across milestones, confirming that the delta matches expected N2P (None-to-Pass, newly added tests) additions and removals. **Test Consistency:** To prevent false penalties, we ensure a negligible Pass-to-Fail (P2F) rate ($\leq 0.026\%$), indicating no unintended regressions are introduced by milestone transitions. Additionally, each milestone is run three times to filter out flaky tests (yielding only 0–16 flaky tests per repo). **Coverage:** Every graded milestone with source changes must contain at least one F2P or N2P test. Milestones lacking test signals are labeled as non-graded maintenance milestones and retained in the execution sequence to preserve code-state and context continuity, but they are excluded from scoring. In the graded evaluation DAG, their incoming and outgoing dependencies are transitively connected to maintain reachability.

## C. SWE-Milestone: Dataset and Construction Details

This appendix collects the construction artifacts of SWE-Milestone: the software requirements specifications, the unified evaluation configuration shared across all agents, per-repository dataset statistics, and the full Milestone DAG visualizations.

### C.1. Software Requirements Specifications

This appendix details the design, structural format, verification protocol, and refinement statistics of the SRS that serves as the task instruction for every milestone in SWE-Milestone.

**Design Principles.**    Each milestone's SRS is written as a *minimally solvable* behavior- and contract-level specification rather than a patch-reconstruction hint. The SRS deliberately avoids line-level edits, patch-like instructions, and identifiers such as file paths or function names unless those identifiers are themselves part of the public contract or are essential to making the requirement unambiguous. Three deliberately competing principles govern this trade-off. First, *specify what is required, not how to implement it*: state motivation, behavior, and acceptance criteria without prescribing the implementation. Second, *align with test intent, not test artifacts*: cover the evaluated behaviors without revealing test names, concrete inputs, or asserted values, exposing only public interface constraints when strictly necessary. Third, *ensure solvability*: include the essential formats, function signatures, edge cases, and non-inferable conventions needed to derive a correct implementation.

**SRS Structure.**    Each milestone's SRS is organized as a sequence of *Feature Requirements*, each containing three fields: a *Problem* statement describing the symptoms and context that motivate the change, a *Requirements* block stating the intended functionality and constraints in an implementation-agnostic way, and an *Acceptance* clause listing observable pass criteria without referencing test artifacts. Together, the three fields bound the implementation space while leaving the agent free to choose any compliant solution. Figure 7 shows the Overview and two illustrative Feature Requirements from the SRS for milestone M001 of `zeromicro/go-zero`, with the remaining requirements omitted for space. Full SRS instances for all milestones are released with the benchmark dataset.

**Human-in-the-Loop Verification.**    The reverse-engineered SRS is refined through an agent-assisted, human-led verification loop. Human annotators iterate over three checks. In *static verification*, annotators read each SRS against the three design principles and revise any obvious violations until the document is internally consistent. In *independent dry-run verification*, annotators dispatch a strong coding agent to attempt each milestone in isolation using only the refined SRS,

---

**Milestone M001 SRS (`zeromicro/go-zero`, v1.6.0→v1.9.3):** *go-redis v9 Upgrade with API Modernization*

**Overview.** This milestone upgrades the go-redis dependency from v8 to v9 across the Redis module (`core/stores/redis`, `core/iox`, `zrpc/internal/clientinterceptors`), requiring a new package import path, migration from callback-based to middleware-based hooks, error handling via `errors.Is()`/`errors.As()`, and updated sorted-set API signatures.

- - - - - - - - - - - - - - - - - - - - - - - - - - - - - - - - - - - - - - - - - - - - - - - - - -

**FR1: Migrate go-redis Import Path**
**Problem.** The Redis client library import path has changed from `github.com/go-redis/redis/v8` to `github.com/redis/go-redis/v9`, causing compilation failures.
**Requirements.**

- Update all import statements referencing the old go-redis v8 package to use the new v9 package path.
- Ensure all Redis-related source files compile with the new import path.
- Maintain the `red` alias convention for the imported package.

**Acceptance.**

- When building the project, no import errors occur for the Redis package.
- All files in the redis module successfully import from the v9 package location.

- - - - - - - - - - - - - - - - - - - - - - - - - - - - - - - - - - - - - - - - - - - - - - - - - -

**FR2: Implement New Hook Interface Pattern**
**Problem.** The go-redis v9 library replaced the callback-based hook interface (`BeforeProcess`/`AfterProcess` methods) with a middleware-style hook interface (`ProcessHook`/`ProcessPipelineHook` functions that wrap the next handler). Existing hook implementations fail to compile.
**Requirements.**

- Implement the `DialHook(next red.DialHook) red.DialHook` method that passes through to the next handler.
- Replace `BeforeProcess`/`AfterProcess` with a single `ProcessHook(next red.ProcessHook) red.ProcessHook` that captures the start time, starts the tracing span, invokes the next handler, ends the span, and records metrics and slow query logs.
- Replace `BeforeProcessPipeline`/`AfterProcessPipeline` with a single `ProcessPipelineHook` method with equivalent behavior for pipeline operations.
- Remove the context-based start-time storage mechanism used by the old hook pattern.
- Update the `startSpan` helper to return both the context and an `endSpan` closure.

**Acceptance.**

- When a Redis command is executed, the tracing span is properly created and completed with correct status.
- When a Redis command exceeds the slow threshold, the slow query is logged.
- When a Redis command fails, the error is properly recorded in the span and metrics.
- When a pipeline of commands is executed, the combined duration is measured and logged appropriately.

- - - - - - - - - - - - - - - - - - - - - - - - - - - - - - - - - - - - - - - - - - - - - - - - - -

. . .

- - - - - - - - - - - - - - - - - - - - - - - - - - - - - - - - - - - - - - - - - - - - - - - - - -

**Environment Dependency Changes.** The SRS additionally lists the Go package upgrades and additions required by this milestone (e.g., `github.com/redis/go-redis/v9` v9.4.0 added, `github.com/go-redis/redis/v8` removed, along with updates to roughly forty transitive dependencies). The full dependency list is included with each milestone's SRS in the released dataset.

*Figure 7.* Excerpt of the SRS for milestone M001 of the `zeromicro/go-zero` repository (v1.6.0→v1.9.3) in SWE-Milestone, showing the Overview, two of the seven Feature Requirements (FR1 and FR2), and the trailing Environment Dependency section. Each Feature Requirement specifies a **Problem**, a **Requirements** block, and an **Acceptance** clause. Full SRS instances for all milestones are available at https://huggingface.co/datasets/DeepCommit-ai/SWE-Milestone-data.

then inspect the failed Fail-to-Pass tests and add the minimal non-inferable conventions required for solvability. After several iterations, state-of-the-art models typically achieve $80\%$ to $90\%$ on independent runs, in line with the per-repository independent-task scores reported in Section 5.2. This indicates that residual unsolvability is small. In *continuous dry-run verification*, annotators replay each repository's DAG end-to-end to surface missing cross-milestone dependencies and SRS gaps that only emerge under sustained execution. A final end-to-end review by an independent senior expert closes the loop.

**Refinement Outcomes.** The verification loop converges quickly. Across SWE-Milestone, $71\%$ of milestones underwent at least one revision, distributed across four categories: removing implementation leakage ($39\%$), removing test leakage ($31\%$), tightening inaccurate or underspecified requirements ($25\%$), and adding missing functional requirements ($5\%$). The convergence is also visible in evaluation impact. The first refinement round shifts per-milestone scores by approximately 5 percentage points, whereas the second round shifts them by at most 2 percentage points, indicating that residual SRS noise has limited effect on the relative ranking of agents.

### C.2. Evaluation Configuration Details

We evaluate multiple model-agent configurations across 4 agent frameworks:

- **Claude Code** (v2.1.50)[1] is Anthropic's terminal-based agent that operates with full shell access and autonomous multi-step planning. We pair it with `Claude Opus 4.5`, `Claude Sonnet 4.5`, `Claude Opus 4.6`, and `Claude Sonnet 4.6`, all supporting a 200K-token context window. Claude Code triggers automatic context compaction at approximately 80% of the context window ($\sim$160K tokens), summarizing prior conversation history via server-side compression while preserving key context.

- **Codex CLI** (v0.105.0)[2] is OpenAI's open-source CLI agent designed for code generation and editing tasks. We pair it with `GPT 5.2`, `GPT 5.2-Codex`, and `GPT 5.3-Codex`, all configured with xhigh reasoning effort and operating with a 272K-token context window. Codex CLI triggers auto-compaction at approximately 90% of context capacity ($\sim$245K tokens). The underlying model is natively trained for multi-context-window operation, automatically summarizing the session and creating a fresh context window while preserving recent messages alongside the summary.

- **Gemini CLI** (v0.29.5)[3] is Google's command-line agent for interacting with Gemini models in development workflows. We pair it with `Gemini 3 Pro`, `Gemini 3.1 Pro`, and `Gemini 3 Flash`, all supporting a 1M-token context window. Gemini CLI triggers context compression at approximately 50% of the context window. The compression mechanism invokes a specialized summarizer that distills the conversation into a structured snapshot preserving the overall goal, key knowledge, file system state, and the agent's current plan.

- **OpenHands** (v1.2.1)[4] (Wang et al., 2025) is an open-source platform for autonomous software development agents. Unlike the provider-specific frameworks above, OpenHands serves as a model-agnostic harness, enabling cross-provider evaluation under a unified agent architecture. We pair it with `Claude Opus 4.6`, `GPT 5.3-Codex`, `Gemini 3 Flash`, `Kimi K2.5`, and `MiniMax M2.5`. OpenHands uses an `LLMSummarizingCondenser` that triggers compression when total tokens surpass a model-dependent threshold (500K for `Gemini 3 Flash`, `Gemini 3 Pro`, and `Gemini 3.1 Pro`, and 160K for all others), preserving the first 4 events (task description and initial context) and using the same underlying model to summarize the remainder.

**Runtime Configuration.** All agents operate within identical sandboxed Docker environments.

**Agent System Prompt.** Under the Continuous Task Evaluation setting, every agent receives the same system prompt (Figure 8) regardless of framework or model. The prompt declares the agent's role as a software engineer, exposes the working directory, source-code paths, task-queue file, and per-milestone SRS directory, lists the four critical constraints (continuous context, in-scope edit boundaries, one-shot tagged submission, and an asynchronously updated streaming queue), and defines the monitor-implement-submit loop in which a `git tag` on the milestone identifier is the sole signal that triggers external evaluation.

---

[1] https://docs.anthropic.com/en/docs/claude-code
[2] https://github.com/openai/codex
[3] https://github.com/google-gemini/gemini-cli
[4] https://github.com/All-Hands-AI/OpenHands

---

**Continuous Task Evaluation Agent System Prompt (`e2e/prompt/v2.md`)**

You are an expert **Software Engineer** working in a continuous integration environment. Your role is to sequentially implement software development tasks from a dynamic queue, maintaining a single continuous context throughout the entire session. You are responsible for writing code, running tests, and managing version control directly.

- - - - - - - - - - - - - - - - - - - - - - - - - - - - - - - - - - - - - - - - - - - - - - - - - - - - - - - - - - - - - - - - - - - - - - - - - - - - - - - - -

**Environment**

- **Working Directory**: `/testbed` (the repository root).
- **Source Code**: {`src_dirs`}.
- **Task Queue File**: `/e2e_workspace/TASK_QUEUE.md` (read-only, updated asynchronously by the system).
- **SRS Directory**: `/e2e_workspace/srs/` (contains {`milestone_id`}_SRS.md files).

- - - - - - - - - - - - - - - - - - - - - - - - - - - - - - - - - - - - - - - - - - - - - - - - - - - - - - - - - - - - - - - - - - - - - - - - - - - - - - - - -

**Critical Constraints**

1. **Continuous Context**: You maintain FULL MEMORY of all previous tasks, decisions, and code changes. Use this knowledge to ensure consistency and avoid regressing previous fixes.
2. **Scope**: Only changes within the Source Code directories ({`src_dirs`}) are validated for the final submission. However, you MAY (and should) modify or add tests to verify your work locally.
3. **One-Shot Submission**: Once you create a submission tag, the task is considered done and removed from the queue. You cannot edit a tagged submission.
4. **Streaming Queue**: The Task Queue is dynamic. New tasks may appear asynchronously as dependencies are satisfied. You must poll it continuously.

- - - - - - - - - - - - - - - - - - - - - - - - - - - - - - - - - - - - - - - - - - - - - - - - - - - - - - - - - - - - - - - - - - - - - - - - - - - - - - - - -

**Workflow.** Follow this continuous loop.
**Step 1: Monitor Task Queue.** Constantly read `/e2e_workspace/TASK_QUEUE.md` to see available tasks. If multiple tasks are available, prioritize them based on the order listed or dependency logic.
**Step 2: Implement Task.** For each task found in the queue:

1. **Read Requirements** from the SRS file at the path shown in TASK_QUEUE.md (format: `/e2e_workspace/srs/{milestone_id}_SRS.md`).
2. **Plan and Implement**: analyze the codebase, plan changes, and modify the code in {`src_dirs`}. **Verify** by running existing tests or by creating new reproduction scripts to ensure that no existing functionality regresses.
3. **Refine** the implementation until satisfied.

**Step 3: Finalize and Submit.** When the implementation is complete and verified:

1. **Commit changes** with `git add {src_dirs}` followed by `git commit -m "Implement {milestone_id}"`.
2. **Tag for submission** via `git tag agent-impl-{milestone_id}`. This is the ONLY signal that the task is complete, and tagging immediately triggers the external evaluator and updates the Task Queue.

**Step 4: Loop.** After tagging, IMMEDIATELY re-read `/e2e_workspace/TASK_QUEUE.md` and repeat Step 2 for any new or remaining tasks. Leverage your memory of previous tasks to handle integration points and shared components effectively.
**Exit Condition.** Only stop when `/e2e_workspace/TASK_QUEUE.md` shows "`(No tasks currently available)`" AND you have completed processing all previously claimed tasks.

*Figure 8.* System prompt provided to every agent under the Continuous Task Evaluation setting in SWE-Milestone. The prompt is framework-agnostic and is loaded as the agent's system message before the first SRS task is read. Variable placeholders such as {`src_dirs`} and {`milestone_id`} are filled in per repository and milestone at runtime.

*Table 3.* Detailed per-repository statistics for SWE-Milestone. HumanDT and HumanWT are average human-estimated development time and SRS writing time, respectively. #M counts graded milestones; the 3 non-graded context milestones are excluded.

| | Codebase | | Release Range | | | Milestone DAG | | |
|---|---|---|---|---|---|---|---|---|
| Org/Repo | #Files | #LoC | Start | End | ΔLoC | #M | #Deps | LoC CV |
| zeromicro/go-zero | 1,021 | 110K | v1.6.0 | v1.9.3 | 6,403 | 23 | 25 | 1.29 |
| apache/dubbo | 4,279 | 350K | 3.3.3 | 3.3.6 | 4,154 | 12 | 9 | 0.76 |
| BurntSushi/ripgrep | 159 | 48K | 14.1.1 | 15.0.0 | 1,474 | 11 | 12 | 0.83 |
| nushell/nushell | 1,727 | 264K | 0.106.0 | 0.108.0 | 15,520 | 13 | 28 | 1.10 |
| element-hq/element-web | 2,430 | 476K | v1.11.95 | v1.11.97 | 7,657 | 18 | 12 | 0.87 |
| navidrome/navidrome | 1,110 | 144K | v0.57.0 | v0.58.0 | 5,900 | 9 | 9 | 1.02 |
| scikit-learn/scikit-learn | 1,314 | 280K | 1.5.2 | 1.6.0 | 7,372 | 12 | 14 | 0.84 |
| **Total/Average** | **12,040** | **1.67M** | — | — | **48,480** | 98 | 109 | 0.96 |

| | Fix Patches | | | SRS | | Unit Tests | |
|---|---|---|---|---|---|---|---|
| Org/Repo | Avg. LoC | Avg. #F | HumanDT | Avg. #W | HumanWT | F2P | P2P |
| zeromicro/go-zero | 278 | 10.2 | 4h-1d | 1,330 | 4h+ | 2.2 | 1,812 |
| apache/dubbo | 346 | 10.8 | 1-3d | 1,138 | 1-2h | 1.1 | 6,924 |
| BurntSushi/ripgrep | 134 | 5.5 | 1-3d | 879 | 30m-1h | 1.3 | 1,057 |
| nushell/nushell | 1,268 | 63.3 | 1-3d | 1,528 | 1-2h | 7.6 | 4,736 |
| element-hq/element-web | 445 | 27.2 | 1-3d | 1,546 | 2-4h | 7.1 | 5,235 |
| navidrome/navidrome | 656 | 13.2 | 1-3d | 1,954 | 30m-1h | 3.0 | 1,452 |
| scikit-learn/scikit-learn | 1,167 | 58.6 | 1-3d | 1,580 | 30m-1h | 97.2 | 22,308 |
| **Total/Average** | 570 | 27.4 | 1-3d | 1,348 | 2-4h | 17.1 | 6,218 |

## C.3. Detailed Dataset Statistics

Table 3 presents comprehensive per-repository statistics for SWE-Milestone. The table is organized into six major categories:

**Repository.** Basic repository information including the organization/repository name on GitHub, the number of source files (#Files), and total lines of code (#LoC) at the end version of the analyzed range.

**Release Range.** The version span analyzed by DeepCommit, specified by start and end tags, along with the delta lines of code (ΔLoC) representing the total code changes between these versions.

**Milestone DAG.** Statistics about the extracted milestone directed acyclic graph: the number of graded milestones (#M; the 3 non-graded context milestones are excluded), inter-milestone dependencies (#Deps), and the coefficient of variation (CV) of patch LoC across milestones. A higher CV indicates greater diversity in milestone complexity within the repository.

**Fix Patches.** Metrics characterizing the gold patches: average lines of code (Avg. LoC), average number of files modified (Avg. #F), and estimated human development time (HumanDT) required to implement each milestone.

**SRS.** Software requirements specification statistics: average word count (Avg. #W) measuring specification length, and estimated human writing time (HumanWT) for authoring equivalent specifications. SRS data is available for all 7 repositories.

**Unit Tests.** Test suite metrics: average Fail-to-Pass (F2P) tests that verify new functionality, and average Pass-to-Pass (P2P) tests ensuring backward compatibility. Notably, scikit-learn exhibits the highest F2P count (97.2) due to its comprehensive test coverage requirements.

## C.4. Milestone DAG Visualizations

Figures 9 to 12 present the full Milestone DAGs for all seven repositories in SWE-Milestone. Each node is a card-style box representing a milestone, containing (top to bottom): a short ID, a descriptive title, summary statistics (number of commits,

lines of code changed, and number of fail-to-pass tests), and one or more category tags. The border color of each node reflects its primary category: **Feature** (blue), **Bugfix** (red), **Refactor** (orange), **Enhance** (teal), or **Chore** (gray). Milestones belonging to multiple categories display all applicable tags. Gray nodes with dashed borders indicate non-graded milestones (3 in total: 2 in ripgrep, 1 in dubbo), which are included in the execution sequence for context continuity but excluded from scoring. The benchmark therefore contains 98 graded milestones and 3 non-graded milestones across all repositories (101 total). Solid red edges denote strong dependencies (upstream removal causes build or test failures downstream), while dashed gray edges denote weak dependencies (functionality degrades gracefully without the upstream milestone). Orange dashed edges represent additional dependencies inferred during the DAG refinement stage. The DAGs are laid out top-to-bottom from upstream prerequisites to downstream dependents. Milestones with no dependency edges are grouped in a dashed "Independent Milestones" box at the bottom of each DAG.

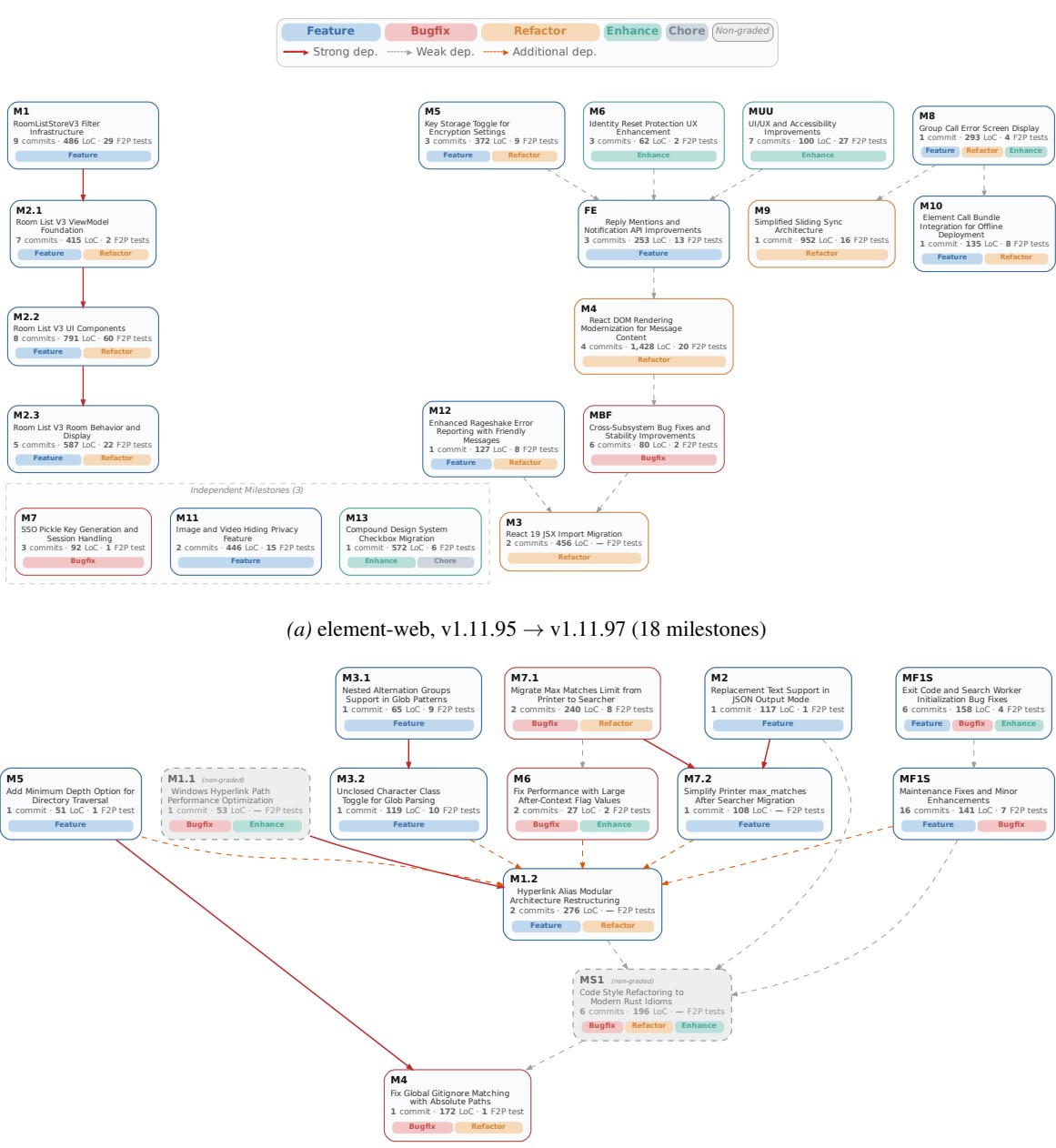

*(a)* element-web, v1.11.95 → v1.11.97 (18 milestones)

*(b)* ripgrep, 14.1.1 → 15.0.0 (11 graded + 2 non-graded milestones)

*Figure 9.* Milestone DAGs for SWE-Milestone repositories (Part 1/4).

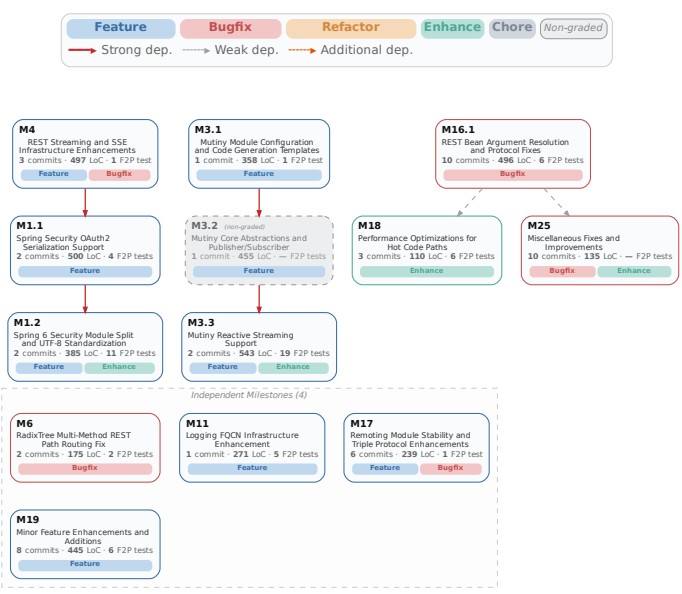

*(a)* dubbo, 3.3.3 → 3.3.6 (12 graded + 1 non-graded milestones)

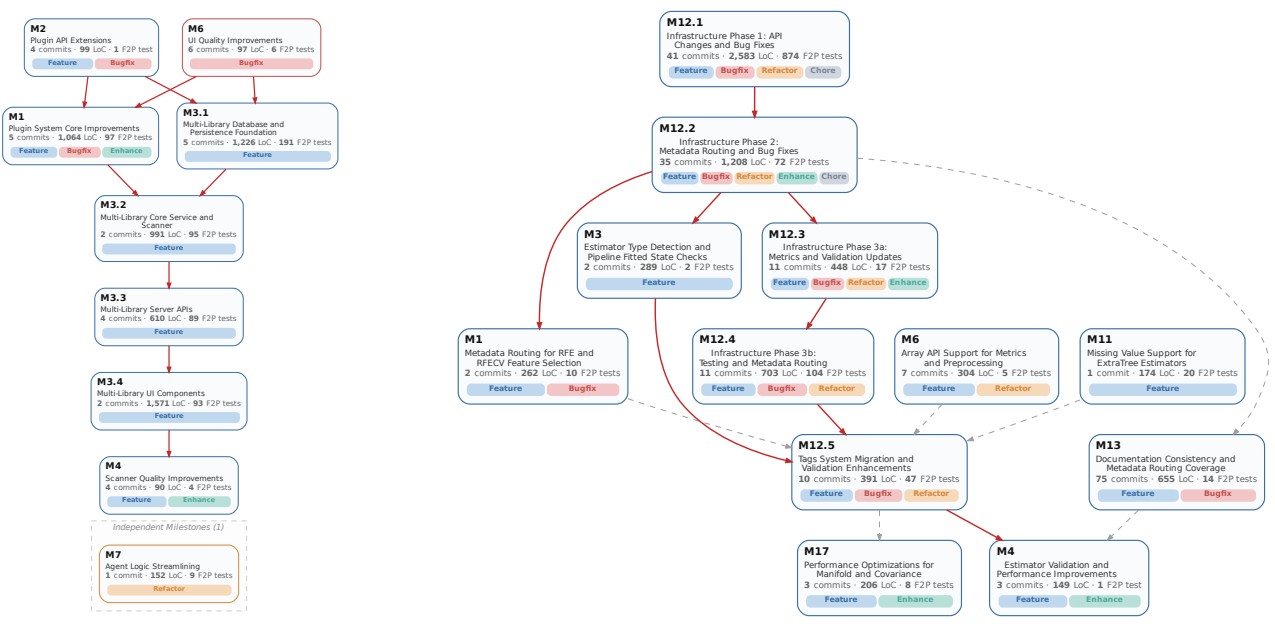

*(b)* navidrome, v0.57.0 → v0.58.0 (9 milestones)

*(c)* scikit-learn, 1.5.2 → 1.6.0 (12 milestones)

*Figure 10.* Milestone DAGs for SWE-Milestone repositories (Part 2/4).

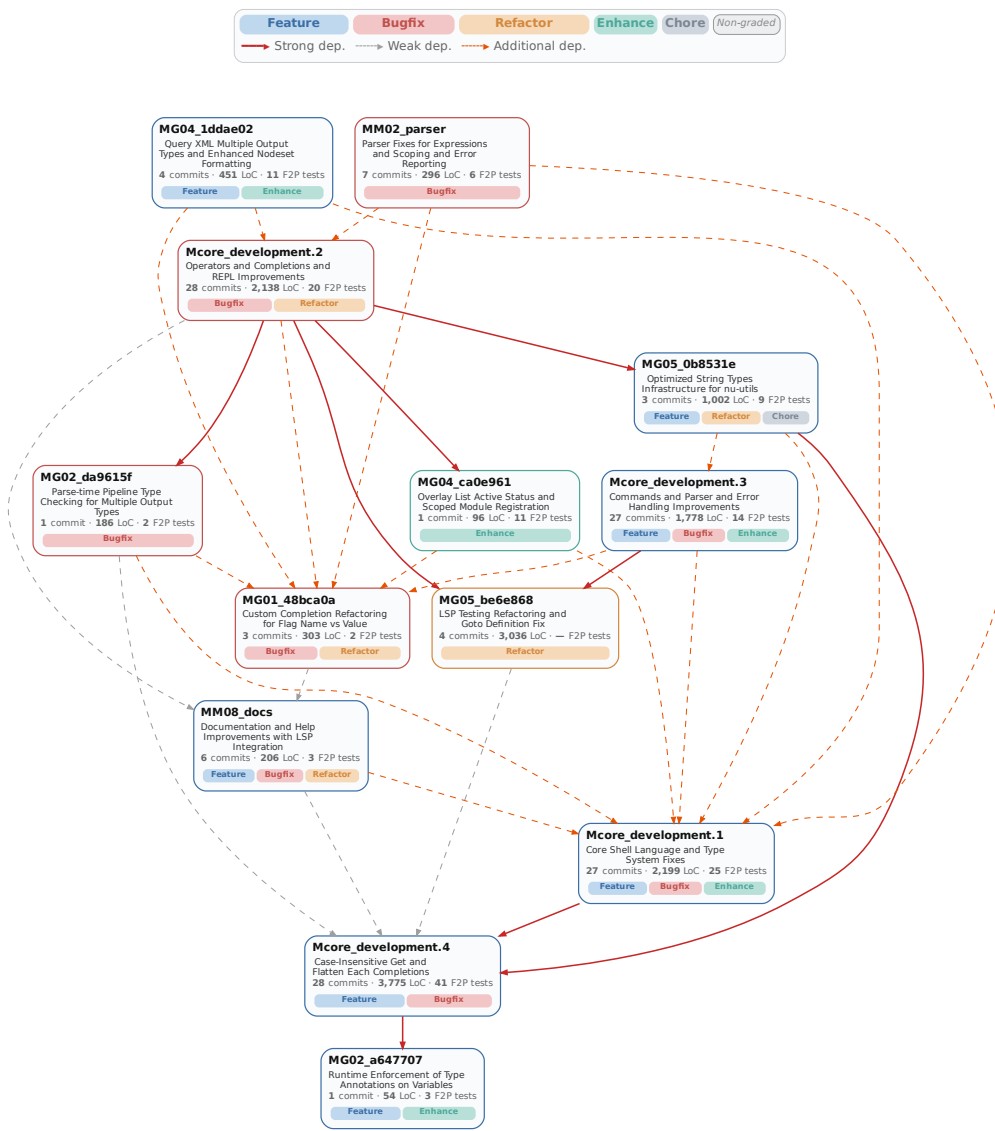

*(a)* nushell, 0.106.0 → 0.108.0 (13 milestones)

*Figure 11.* Milestone DAGs for SWE-Milestone repositories (Part 3/4).

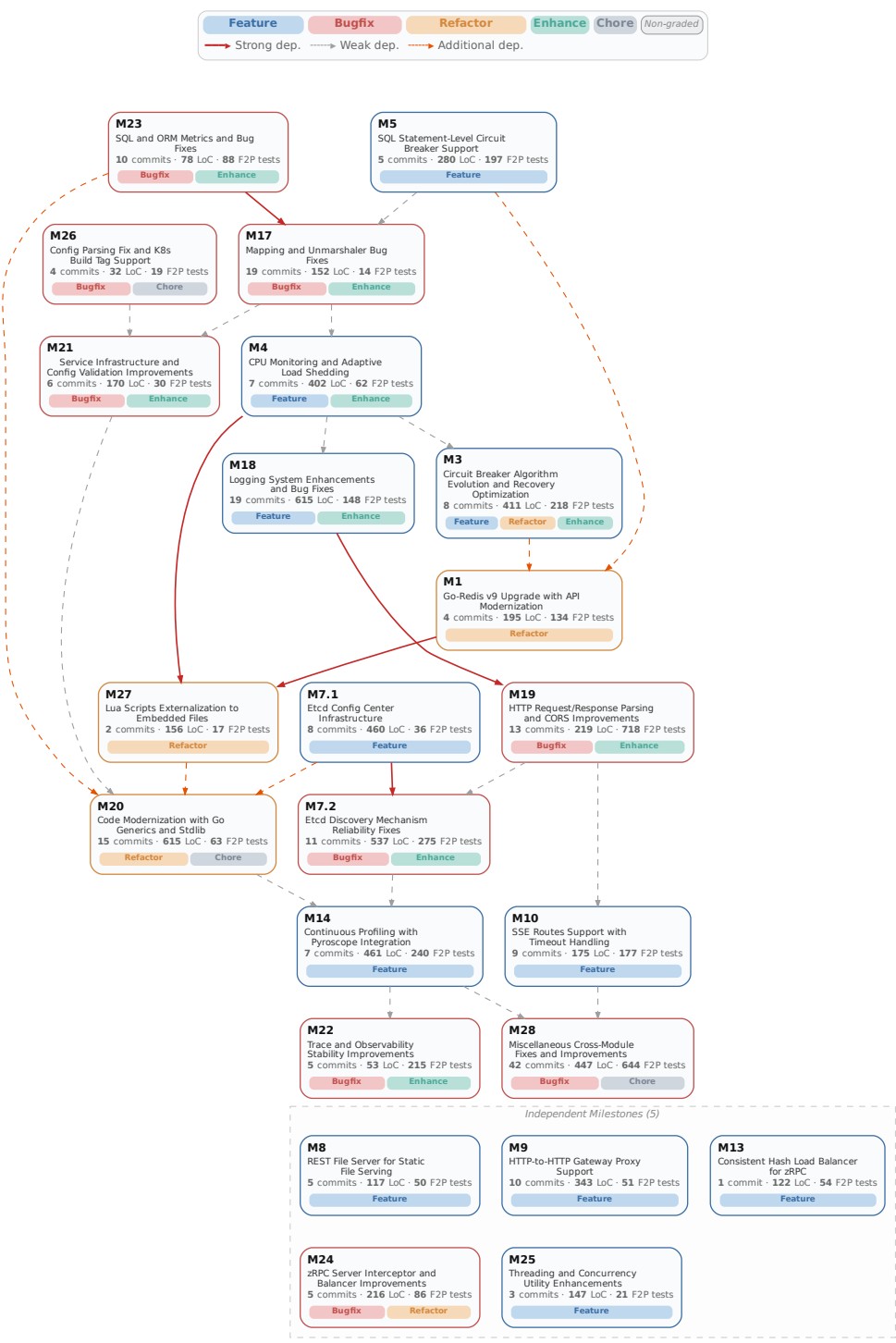

*(a)* go-zero, v1.6.0 → v1.9.3 (23 milestones)

*Figure 12.* Milestone DAGs for SWE-Milestone repositories (Part 4/4).

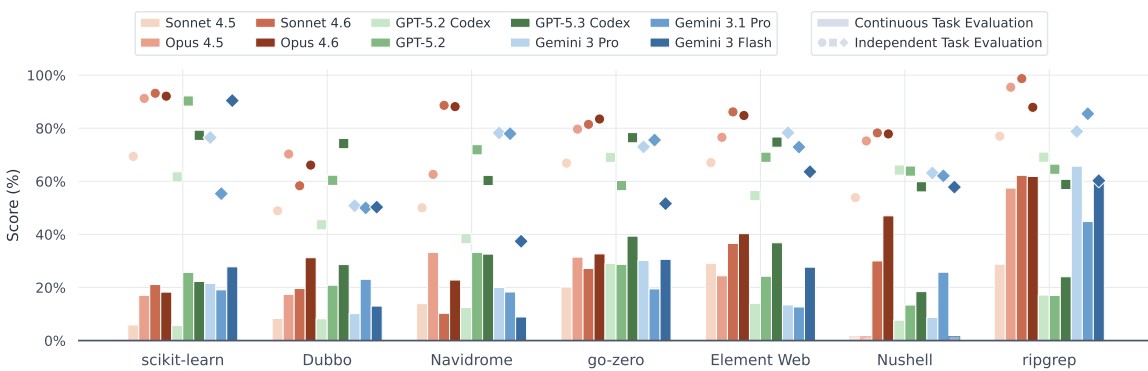

*Figure 13.* Per-repository score comparison under two evaluation modes. High independent-task performance across all repositories confirms that milestones are individually solvable.

## D. Extended Experimental Analysis and Discussion

This appendix expands the experimental analyses summarized in Section 5, including extended overall performance breakdowns, task complexity and topological effects, the full failure-chain analysis, behavioral telemetry, qualitative agent-code quality patterns, per-model cumulative error trajectories, and a comparison case study against human-annotated DAGs.

### D.1. Extended Overall Performance Analysis

This appendix expands the overall-performance discussion in Section 5.2 with per-model-family comparisons, the cost–score trade-off, and the per-repository breakdown.

**Model-family generational comparison.** Comparing across model families, clear generational improvements emerge: Claude 4.6 models significantly outperform their 4.5 predecessors, and `GPT 5.3-Codex` substantially improves over both `GPT 5.2-Codex` and `GPT 5.2`. However, the comparison between `GPT 5.2` and `GPT 5.2-Codex` suggests that optimization for isolated coding tasks does not necessarily transfer to long-horizon development in this setting, where sustained codebase maintenance demands broader analytical capabilities. The three Gemini models achieve comparable scores, with `Gemini 3 Flash` matching `Gemini 3 Pro` at one-ninth the cost. `Gemini 3 Pro` uses the fewest turns, possibly indicating insufficient exploration.

**Cost–score trade-off.** Figure 15 shows that higher cost does not uniformly translate into higher performance: `Gemini 3 Pro` exceeds $100 per evolution range yet scores below `Claude Opus 4.6` ($88), and `Claude Sonnet 4.6` ($69) trails `Claude Opus 4.6` by 6.7 points despite a similar cost tier. On the Pareto frontier, `Gemini 3 Flash` ($12, 24.2%) and `GPT 5.3-Codex` ($25, 28.9%) offer the best cost-effectiveness, achieving competitive scores at a fraction of the cost of top-performing models. OpenHands trials exhibit notably longer execution times (e.g., 18.75 h for `GPT 5.3-Codex`) because its runner permits up to 3,000 iterations per milestone with automatic session resumption, allowing the agent to retry extensively when stuck. This additional compute does not consistently improve scores: `Claude Code` with `Claude Opus 4.6` achieves a comparable score in under 4 hours.

**Per-repository comparison.** Figure 13 compares per-repository performance under both evaluation modes. The continuous–independent gap varies significantly by repository, with `scikit-learn` exhibiting the largest degradation: `Claude Sonnet 4.6` achieves 93.2% independently but only 21.1% under continuous evaluation.

### D.2. Extended Task Complexity and Topological Effects

Figure 14 examines how milestone characteristics correlate with agent performance. Gold patch LoC, a traditional measure of task complexity, shows a clear monotonic relationship. Larger patches require more code changes and yield lower scores across all models. SRS (Software Requirements Specification) word count, however, exhibits a non-monotonic pattern, with a clear sweet spot for specifications of moderate length (around 500 to 1500 words). When specifications are concise, agents

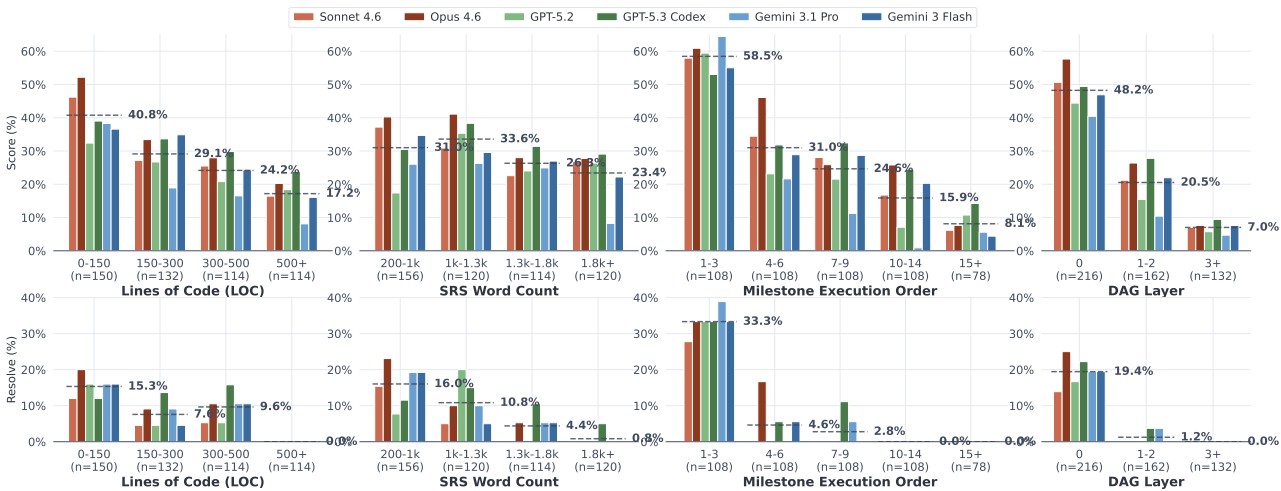

*Figure 14.* Task complexity effects on Score (top) and Resolve Rate (bottom). Milestones are binned by code size, specification length, execution order, and DAG layer. Dashed lines show per-bin averages across all models.

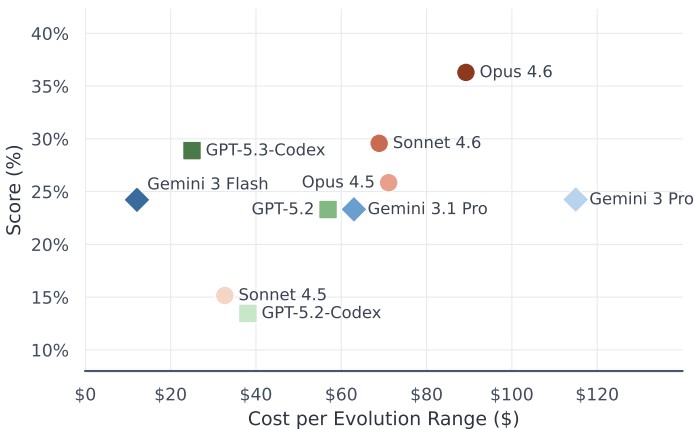

*Figure 15.* Overall Score vs. cost trade-off across all repositories.

must autonomously locate relevant context from the repository, increasing exploration burden. When specifications are verbose, the sheer volume of requirements increases implementation workload. Milestones with moderate-length SRSs achieve the highest accuracy, suggesting that task difficulty depends not only on implementation effort but also on the cost of information acquisition.

Beyond these static factors, the continuous evaluation setting introduces structural complexity unique to SWE-Milestone. Both the milestone execution order and the DAG topological layer show statistically significant negative correlations with the score. Later milestones and deeper topological layers consistently yield lower performance. This reflects the compounding effect of upstream errors, as agents must build upon their own (potentially flawed) prior work. These topological factors are absent in independent evaluation and represent the distinctive challenge of long-horizon software evolution. The Resolve Rate (bottom row of Figure 14) makes this effect even starker: it drops drastically beyond the earliest milestones and the shallowest DAG layers, indicating that current agents can only fully resolve milestones that appear early in the sequence or have no upstream dependencies. Once prior errors accumulate, agents may still achieve partial progress (reflected in Score) but rarely produce a completely correct solution.

### D.3. Extended Failure Analysis: Error Generation and Propagation

Understanding *why* agents fail in continuous evaluation is inherently difficult. A single early mistake can trigger cascading test failures across dozens of downstream milestones, making it challenging to disentangle root causes from their propagated

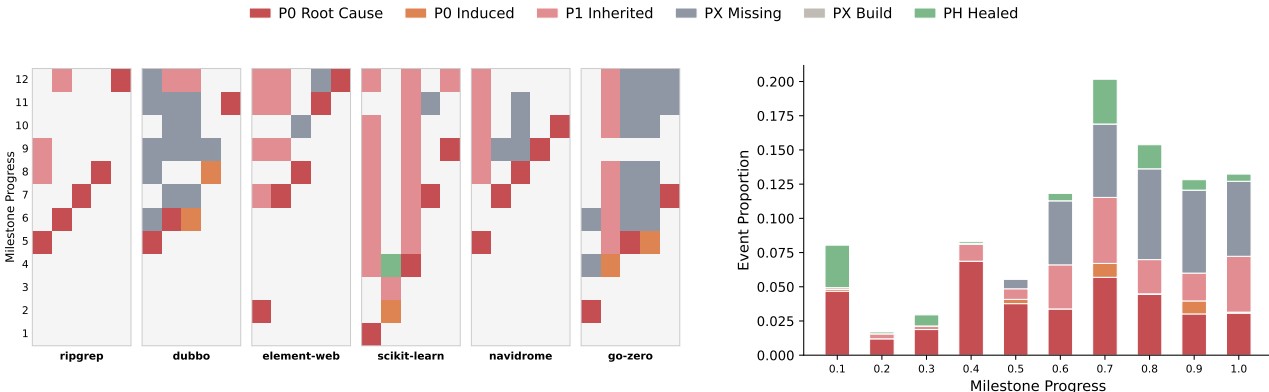

*Figure 16.* Propagation type analysis for `Claude Opus 4.6`. **Left**: Selected error chain patterns across repositories, where each column is an error chain and each row a milestone. **Right**: Distribution of propagation event types across milestone progress bins (averaged over all repositories), showing how inherited failures (P1) and infrastructure effects (PX) increasingly dominate in later stages.

consequences. To enable systematic analysis, we introduce the concept of **error chains**: for each test that transitions from passing to failing during the evolution, we trace its status across all subsequent milestones until it is either healed or the trial ends. This yields a per-test timeline that captures the full lifecycle of an error. We focus on the strongest vendor-framework pairing, `Claude Code` with `Claude Opus 4.6`, to characterize frontier failure mechanisms.

We decompose error chains along two orthogonal dimensions. The first, **Propagation Type**, captures *how* a fault affects downstream milestones. This is determined statistically from evaluation results: we track each test's status across the milestone timeline and classify events as P0 Root Cause (the originating failure), P0 Induced (cross-chain contamination from unrelated changes), P1 Inherited (propagated through dependency), PX Missing (skipped execution), or PH Healed (successfully recovered). Figure 16 (right) shows that propagation events (P1, PX) increasingly dominate in later stages, confirming the compounding degradation observed in Appendix D.2. The left panel visualizes representative error chain patterns across repositories, illustrating how a single root cause event can cascade through the entire remaining evolution.

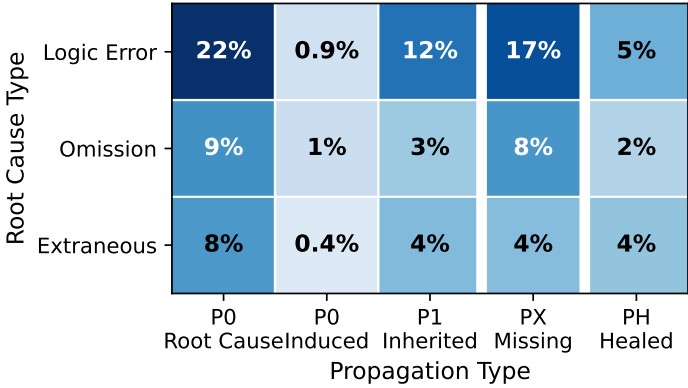

*Figure 17.* Root Cause Type × Propagation Type heatmap. Each cell shows the macro-averaged event proportion across all repositories.

The second dimension, **Root Cause Type**, captures *why* the initial fault originates. Since root cause attribution requires understanding the agent's intent, we employ `Claude Sonnet 4.6` as a reviewer agent that compares the task agent's code changes against the ground-truth patch, the SRS specification, and evaluation artifacts. The reviewer classifies each error chain's root cause into three categories: Logic Error (correct target, buggy implementation), Omission (missing a required component), or Extraneous (unnecessary modifications that break existing functionality). Figure 17 presents the joint distribution of Root Cause Type × Propagation Type. Logic Error is the dominant root cause (∼57% of all error chain events), with its chains exhibiting both the highest inherited propagation (P1, 12%) and the highest proportion of missing test execution (PX Missing, 17%), indicating that buggy implementations frequently prevent downstream tests from running at all.

## D.4. Extended Agent Behavior Analysis

Beyond aggregate scores, we examine how agents allocate effort and manage state when facing accumulating technical debt during long-horizon iterations. By instrumenting tool calls, context usage, and interaction turns, we reveal distinct behavioral patterns.

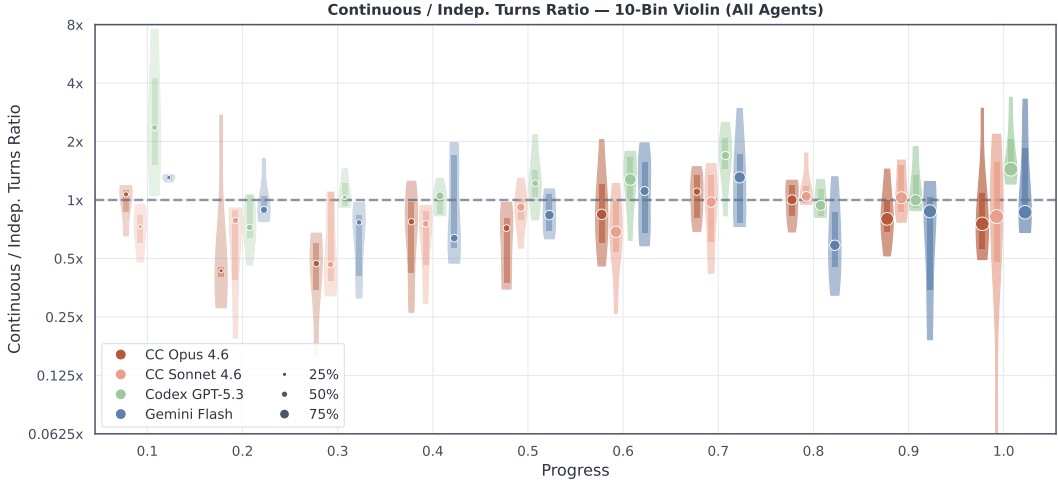

*Figure 18.* Continuous-to-Independent turns ratio across normalized execution progress (10 bins). Values above $1\times$ indicate the agent expends more effort under continuous evaluation than on the same milestone independently. The ratio remains near or below $1\times$ for most of the sequence but rises sharply in the final bin. This reflects increased rework and error-recovery effort as accumulated technical debt compounds.

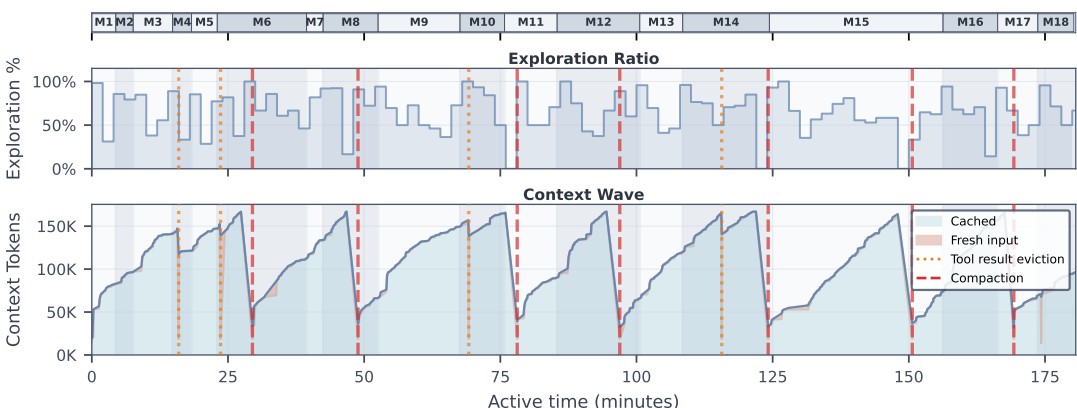

*Figure 19.* Exploration ratio and context wave for `Claude Code` (powered by `Claude Opus 4.6`) on `element-web`. Top plot shows milestone progression (M1 to M18). Middle plot shows per-minute exploration ratio (read/search vs. write/execute). Bottom plot shows total context token usage over active time, with compaction and eviction events marked.

**Effort Fluctuation and Extremes.** As shown in Figure 18, the four representative vendor-framework configurations exhibit a shared trend in their effort allocation (measured by the continuous-to-independent turns ratio). In the initial phase (progress ~0.1), continuous effort is slightly higher than independent effort, as agents must conduct large-scale exploration to build a mental model of the unfamiliar repository. During the middle phase (progress 0.1–0.5), the ratio drops below $1\times$ (with the median falling to ~0.83×): agents successfully reuse their established context, bypassing the redundant exploration required in independent evaluation. However, in the late stage (progress 0.6–0.9), effort rises significantly as accumulating errors demand extensive debugging. Finally, near completion (progress ~1.0), agent behavior diverges sharply. Some agents resort to frantic thrashing, while others prematurely give up. Notably, `GPT 5.3-Codex` demonstrates the most stable effort profile, maintaining consistent variance throughout the project lifecycle.

**Context Stability and Exploration Patterns.** To sustain this fluctuating effort, agents must effectively manage their context. Figure 19 illustrates this using `Claude Code` with `Claude Opus 4.6` as a representative example. The context window shows stable, controllable wave patterns, demonstrating that modern agent frameworks paired with frontier models can effectively support long-horizon programming without catastrophic context overflow. The framework employs two compression strategies: partial compression (evicting specific tool results) and heavy compaction (summarizing extensive histories). Crucially, agent exploration behavior (reading and searching) is tightly coupled with this state management. Exploration surges at the beginning of each new milestone and immediately following major context compaction events, as the agent works to rebuild its mental model.

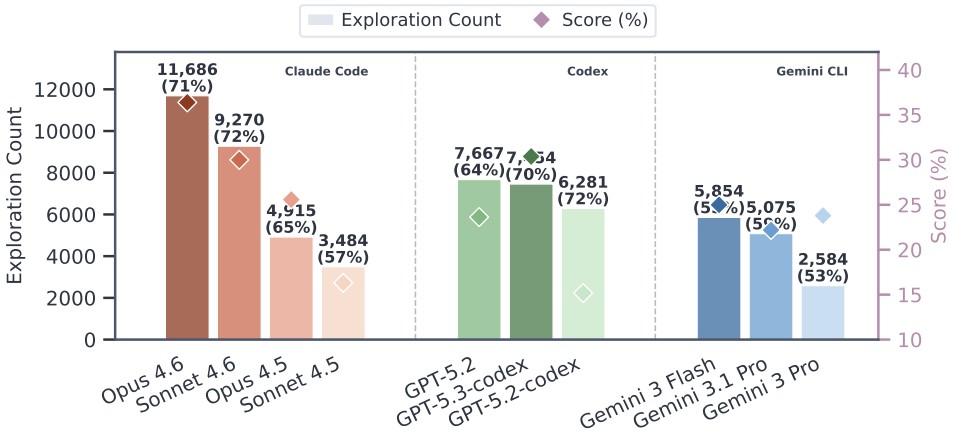

*Figure 20.* Exploration tool-call count aggregated across all repositories, grouped by agent framework and sorted by count within each cluster. Diamond markers show average Score. Higher-performing agents consistently devote greater effort to exploration (e.g., reading and searching).

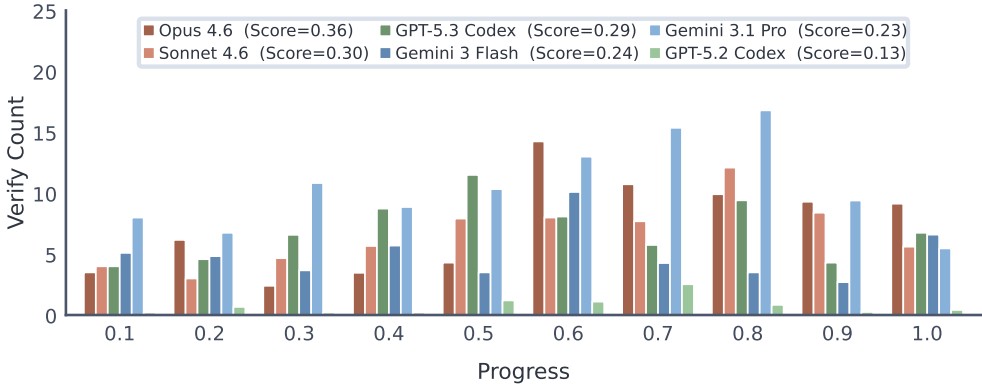

*Figure 21.* Average verification tool-call count per milestone progress bin (10 bins), for representative agent configurations. Verification effort generally increases with progress. It peaks around 70% to 80% completion before declining in the final bin.

**The Impact of Exploration.** Exploration behavior is strongly associated with downstream success. Figure 20 demonstrates that within their respective agent frameworks, models from the same family exhibit a consistent pattern: higher exploration counts correlate with better performance. For instance, `Claude Opus 4.6` and `Claude Sonnet 4.6` hold a distinct advantage because they aggressively dispatch subagents to analyze the codebase, executing over 7,000 exploration commands. Conversely, models like `Gemini 3 Pro` allocate too little effort to reading, indicating that many current models still lack proactive exploration for long-horizon tasks.

**Verification vs. Blind Thrashing.** Alongside exploration, we analyze verification behavior (test execution). Figure 21 shows that average verification effort generally follows an inverted-U shape, increasing as the codebase grows more complex before declining near the end. However, this masks two problematic extremes: `Gemini 3.1 Pro` verifies excessively,

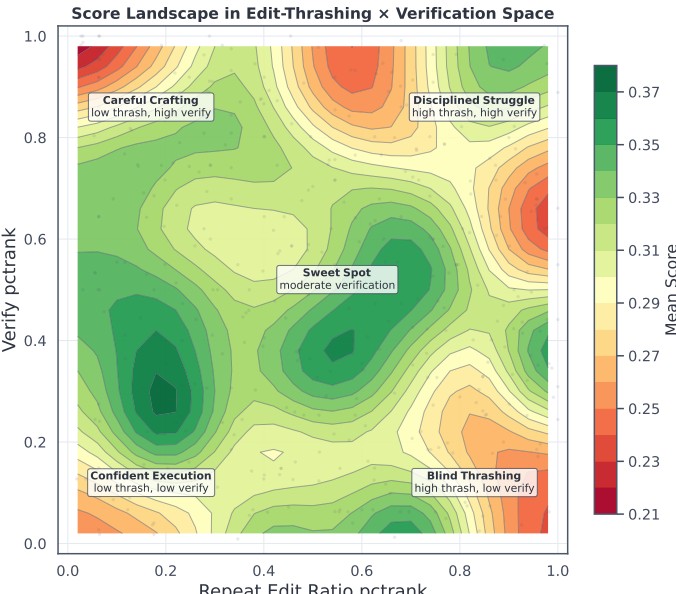

*Figure 22.* Score landscape in the edit-thrashing (repeat edit ratio) vs. verification (test execution frequency) space, aggregated across all repositories. A sweet spot of moderate thrashing and moderate verification yields the highest scores. The blind thrashing quadrant (high thrash, low verify) produces the worst outcomes.

while `GPT 5.2-Codex` rarely verifies at all, and both achieve lower scores. Figure 22 further isolates this dynamic by mapping the score landscape against edit thrashing and verification frequency. A clear sweet spot emerges for moderate, disciplined verification. In contrast, the worst outcomes concentrate in the high-thrash and low-verify quadrant—a blind thrashing trap where agents repeatedly modify the same files without executing tests to guide them, effectively accelerating the snowball effect.

### D.5. Agent Code Quality Analysis

Beyond aggregate metrics, we examine representative cases to understand *how* agent-generated patches differ from human-written patches in structural quality. We identify three recurring quality anti-patterns through qualitative analysis.

**Responsibility Boundary Misplacement.** Figure 23 illustrates a case from `apache/dubbo` (M004) where the agent correctly identifies the logic to implement but places it at the wrong abstraction level. The task requires skipping deserialization for stream parameters in `FallbackArgumentResolver`. The ground truth inserts the check in `resolveValue()`, allowing `accept()` to still claim the parameter, after which the framework injects the real stream instance. The agent instead places the check in `accept()`, causing the terminal resolver to reject the parameter entirely and triggering 3 P2P regressions. The agent's solution is *functionally plausible*—checking `isStream()` before processing is a reasonable heuristic—but violates the resolver chain's structural contract, revealing a quality gap in architectural reasoning.

**Shotgun Fixes.** Figure 24 shows a pattern from `nushell` where the agent replaces targeted validation with overly permissive error suppression. The task requires `break/continue` outside loops to produce compile-time errors. The ground truth adds an `is_in_loop()` guard in `compile_break()` and precisely exempts only the known `catch` block false positive. The agent instead deletes the `is_in_loop()` API entirely, removes all guards from `compile_break()`, and globally suppresses `NotInALoop` errors across all block expressions. Tests still pass because an internal fallback in `push_break()` happens to catch the error—but the SRS-mandated API contract is violated, and the deleted API blocks future evolution of the compiler's context-tracking infrastructure. This pattern exemplifies a broader quality gap: agents achieve test-passing behavior through coarse-grained suppression rather than precise, contract-preserving fixes.

**API Signature Degradation.** Figure 25 demonstrates how a local algorithm choice can silently degrade a public API. In `nushell`'s multi-output type checking, the ground truth iterates per input-output pair, keeping the `expected` field of

`OutputMismatch` as a structured `Type` enum, adds a reusable utility in `ty.rs`, and cleans up obsolete workarounds (7 files modified). The agent groups by unique input types, which forces the `expected` field to become an opaque `String` (losing pattern-matchability), defines only a local helper (leaving 35 lines of duplicate logic elsewhere), and retains stale FIXME workarounds (2 files modified). Both pass all F2P tests, but the agent's patch introduces architectural debt—weaker type signatures, missed deduplication, and retained technical debt—that compounds across milestones.

These cases reveal a common quality pattern: agent-generated patches achieve *superficial correctness* (tests pass) while introducing *structural and maintainability issues*—misplaced abstractions, coarse-grained suppression, and degraded API contracts—that are invisible to automated evaluation but consequential in long-horizon development.

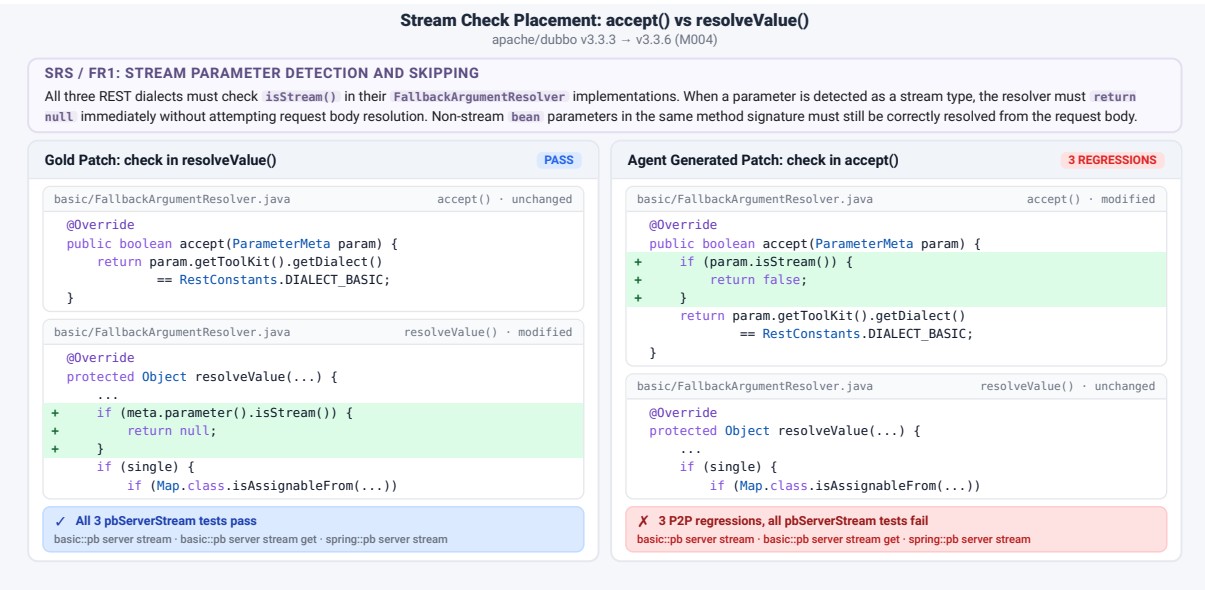

*Figure 23.* Responsibility boundary misplacement in `apache/dubbo` (M004). The ground truth checks stream parameters in `resolveValue()`, preserving the resolver chain contract. The agent checks in `accept()`, ejecting the parameter from the terminal resolver and causing 3 P2P regressions.

## D.6. Cumulative Error Analysis on `element-web`

Figure 26 contrasts continuous and independent evaluation on `element-web` across multiple models. Under continuous evaluation, errors introduced at early milestones, such as regressions and unresolved bugs, propagate through subsequent development stages, producing a pronounced *snowball effect* in which agents must operate on an increasingly unstable codebase. In contrast, `Gemini 3 Flash` under independent evaluation achieves substantially higher performance than all continuous runs, including those of frontier models such as `GPT 5.2` and `Claude Opus 4.5`. This gap illustrates that independent-task evaluation effectively serves as an optimistic upper bound, substantially overstating an agent's ability to sustain coherent software evolution. Together, these results highlight long-horizon codebase maintenance as a central bottleneck: even state-of-the-art agents struggle to control error accumulation and technical debt when development unfolds continuously.

## D.7. Case Study: Human vs. DeepCommit Milestone DAG Construction

We conduct a structured comparison between the Human-annotated Milestone DAG and the DeepCommit DAG for the `scikit-learn` v1.5.2–v1.6.0 release interval (373 commits over 7 months). As shown in Figures 27 and 28, the Human DAG consists of 14 milestones organized into 6 semantic groups with 19 dependency edges, whereas the DeepCommit DAG consists of 12 milestones organized into 6 data-driven groups with 14 dependency edges.

Both DAGs adopt a two-level hierarchy (groups → milestones). However, the Human DAG is structured around manually defined release-planning themes (e.g., framework evolution, backend compatibility, quality assurance), while the Deep-

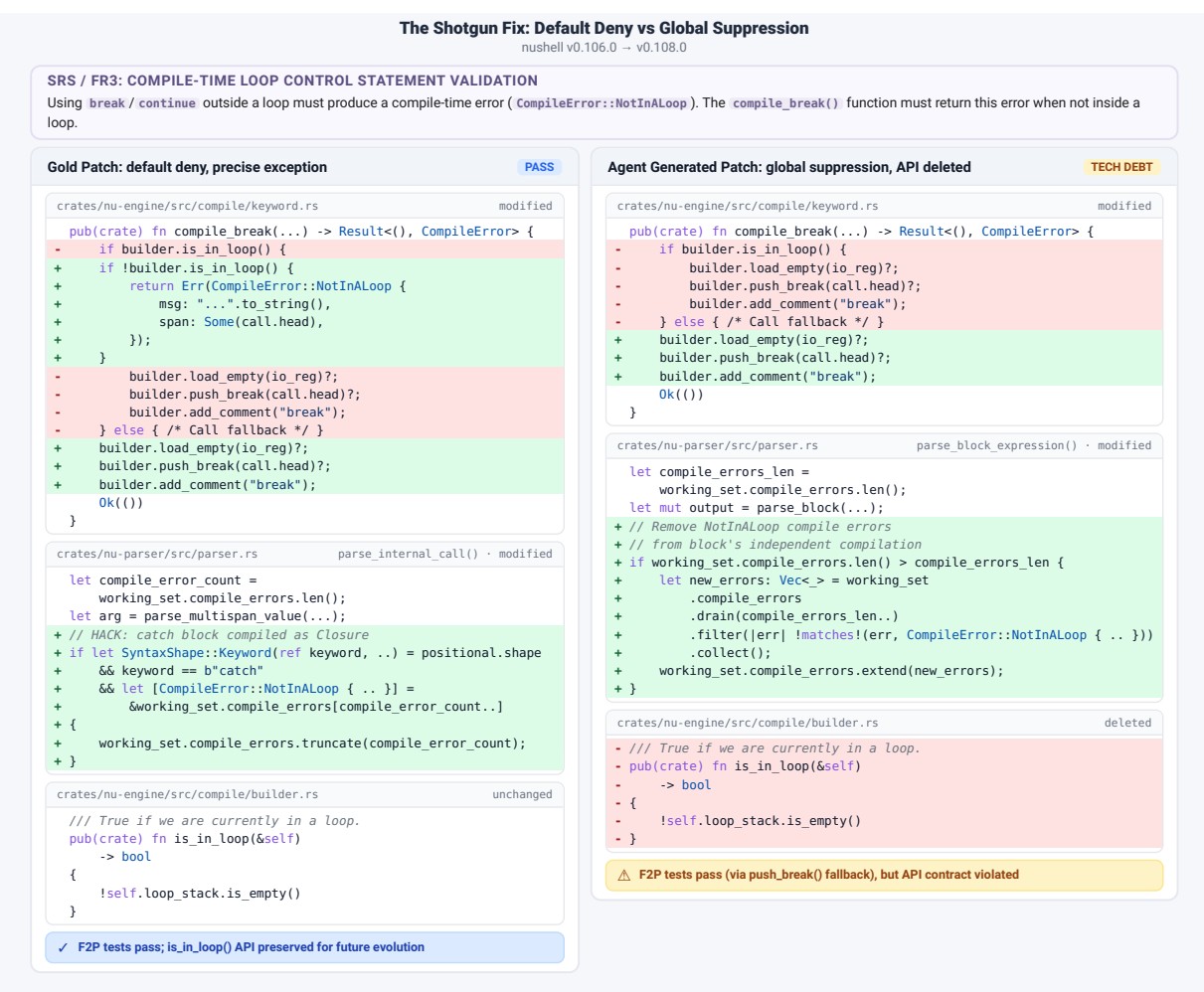

*Figure 24.* Shotgun fix in `nushell`. The gold patch uses a *whitelist* strategy: `compile_break()` returns `NotInALoop` as required by the SRS, and only the known `catch`-block false positive is exempted in `parse_internal_call`. The agent uses a *blacklist* strategy: it suppresses *all* `NotInALoop` errors across every block expression and deletes the `is_in_loop()` API. Tests pass only because `push_break()` has an internal fallback.

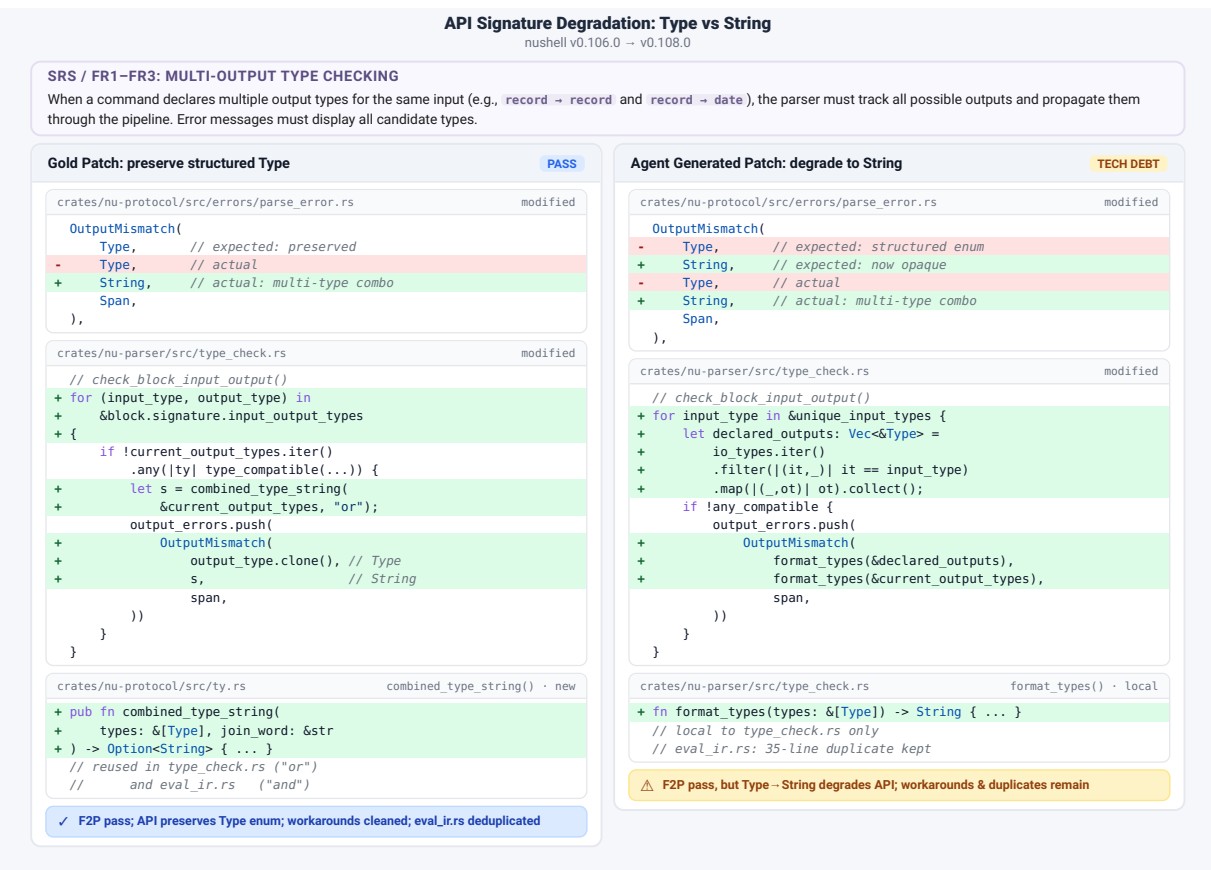

*Figure 25.* API signature degradation in `nushell`'s type checker. The agent's group-by-input algorithm forces `OutputMismatch.expected` from a structured `Type` enum to an opaque `String`, losing pattern-match capability. The agent also omits workaround cleanup and utility deduplication (2 vs. 7 files modified).

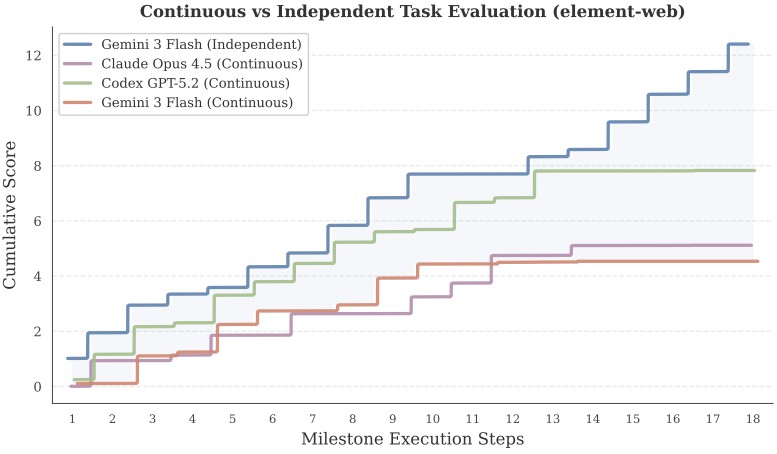

*Figure 26.* Continuous vs. independent evaluation on `element-web`. The shaded area highlights how error accumulation in continuous mode causes a cost-effective model (`Gemini 3 Flash`, independent) to outperform frontier models in continuous mode.

*Table 4.* Structural overview of the DeepCommit Milestone DAG and the Human-annotated Milestone DAG.

| Dimension | Human DAG | DeepCommit DAG |
|---|---|---|
| Leaf milestones | 14 | 12 |
| Groups | 6 | 6 |
| Covered commits | 373 | 201 |
| DAG edges | 19 | 14 |
| Edge types | FUNC (14) + NFR (5) | FUNC (13) + NFR (1) |
| Strong edges | 2 | 8 |
| Weak edges | 17 | 6 |

Commit DAG reflects phases and clusters induced from commit-level dependency topology. In addition, the Human DAG distinguishes functional and process-level dependencies with explicit strength annotations, whereas the DeepCommit DAG encodes dependencies derived from structural signals in the commit graph.

This case study is not intended as a performance comparison. Rather, it examines how distinct construction principles—semantic curation versus topology-driven aggregation—lead to systematically different milestone abstractions and dependency structures.

### D.7.1. COMMIT COVERAGE

The 201 DeepCommit commits form a strict subset of the 373 Human commits. The 172 filtered commits result from the pipeline's source-file filtering and F2P test requirements, which preferentially exclude Human milestones whose commits have low inter-dependency, since such commits tend to modify isolated files and fail to form strongly connected components in the dependency-induced commit DAG. Among the most affected categories, 73% of build system modernization commits and 78% of Array API interface adaptation commits are excluded, followed by CI/release engineering (56%) and small independent bug fixes (39%). This strategy retains a tighter, more interconnected subgraph suited for benchmark construction, but omits process-level patterns such as "code first, document later" that the Human DAG captures via NFR edges.

### D.7.2. CLUSTERING DIVERGENCE AND STRUCTURAL DIFFERENCES

Although both DAGs have 6 groups, their clustering methods are fundamentally different, producing an Adjusted Rand Index (ARI) of only 0.538 over the 201 shared commits, indicating moderate agreement despite identical group counts.

**Top-down semantics vs. bottom-up topology.** The human annotator reads the release notes[5] and resolved issues[6] to define semantic cluster centers (e.g., "Estimator Tags System Redesign" → M1.2, "Deprecations and Removals" → M3.2), then assigns commits by *developer intent*. DeepCommit instead constructs a commit-level DAG from code dependencies and partitions it via seed discovery and consolidation. The resulting chain M1.1→M1.2→M2.1→M2.2→M2.3 is a contiguous topological ordering.

The contingency matrix (Figure 29) illustrates this difference from both directions. Human M5.1 (Stability Fixes, 37 shared commits) groups bug fixes spanning the entire release window by shared intent, but DeepCommit distributes them across eight milestones (M1.1:13, M1.2:12, M2.1–M2.3:7, etc.) according to which code regions they touch. Conversely, DeepCommit M1.1 (41 commits) clusters code-adjacent early-phase work into a single topological block, drawing from four distinct Human categories: stability fixes (13), deprecation cleanup (9), usability improvements (6), and release engineering (6).

**Group semantics and dependency structure.** The two construction methods yield different group semantics. Human groups reflect *strategic release themes* with high internal coherence (e.g., M1: testing → tags → validation → routing, all under "Framework Evolvability"), whereas DeepCommit groups reflect *topological phases*: M1 ("Framework Evolvability," 76 commits) aggregates license cleanup, metadata routing, and estimator bug fixes—commits that are code-adjacent in time but span multiple developer-facing themes. This difference extends to edges. The Human DAG distinguishes FUNC edges

---

[5]https://scikit-learn.org/stable/auto_examples/release_highlights/plot_release_highlights_1_6_0.html
[6]https://github.com/scikit-learn/scikit-learn/milestone/57?closed=1

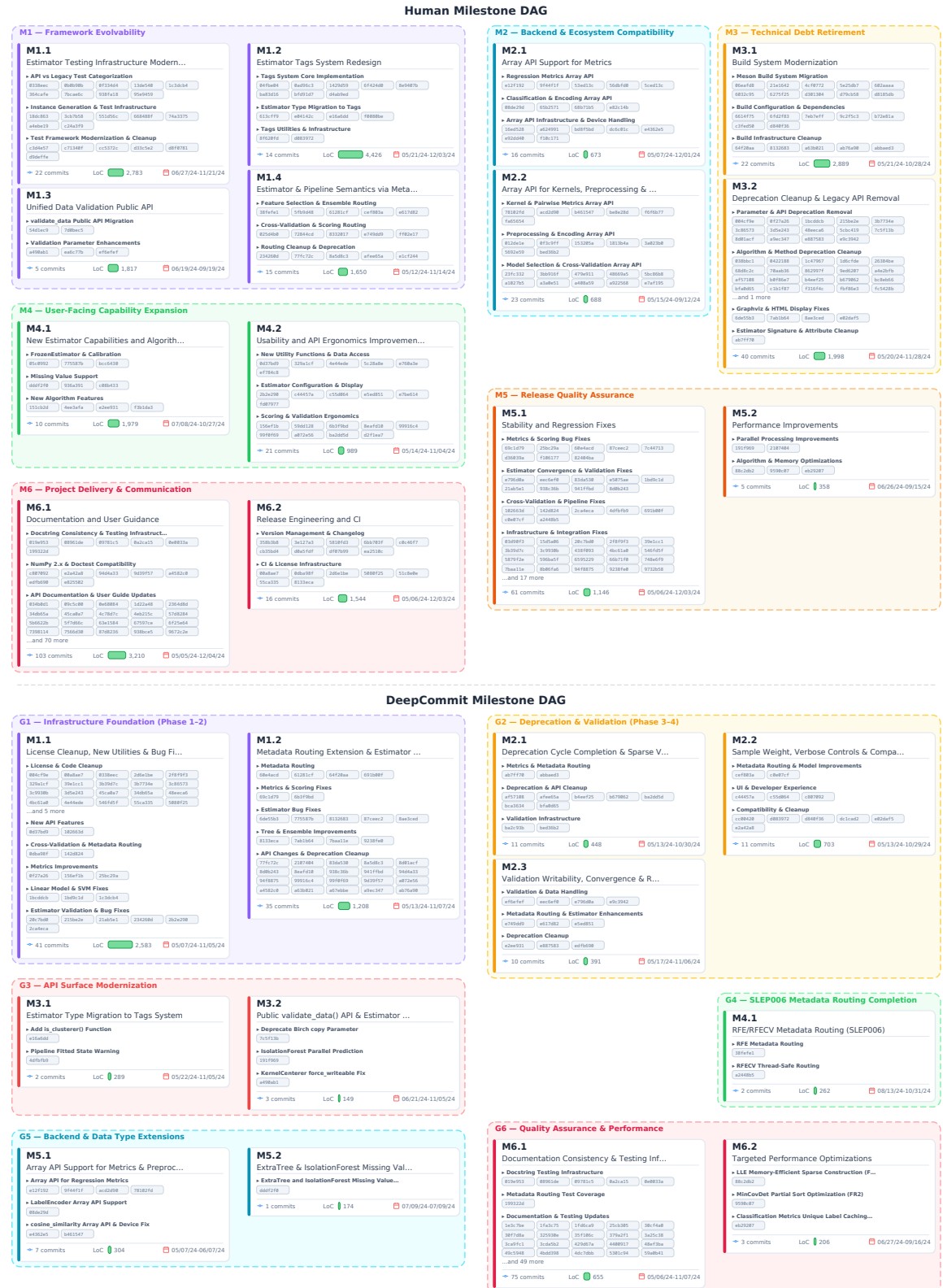

*Figure 27.* Human-annotated and DeepCommit milestone decompositions for `scikit-learn` v1.5.2–v1.6.0. Top: milestones grouped by human analyzers (M1–M6). Bottom: milestones grouped by DeepCommit (G1–G6). Each block summarizes representative commits, commit count, LoC, and time span.

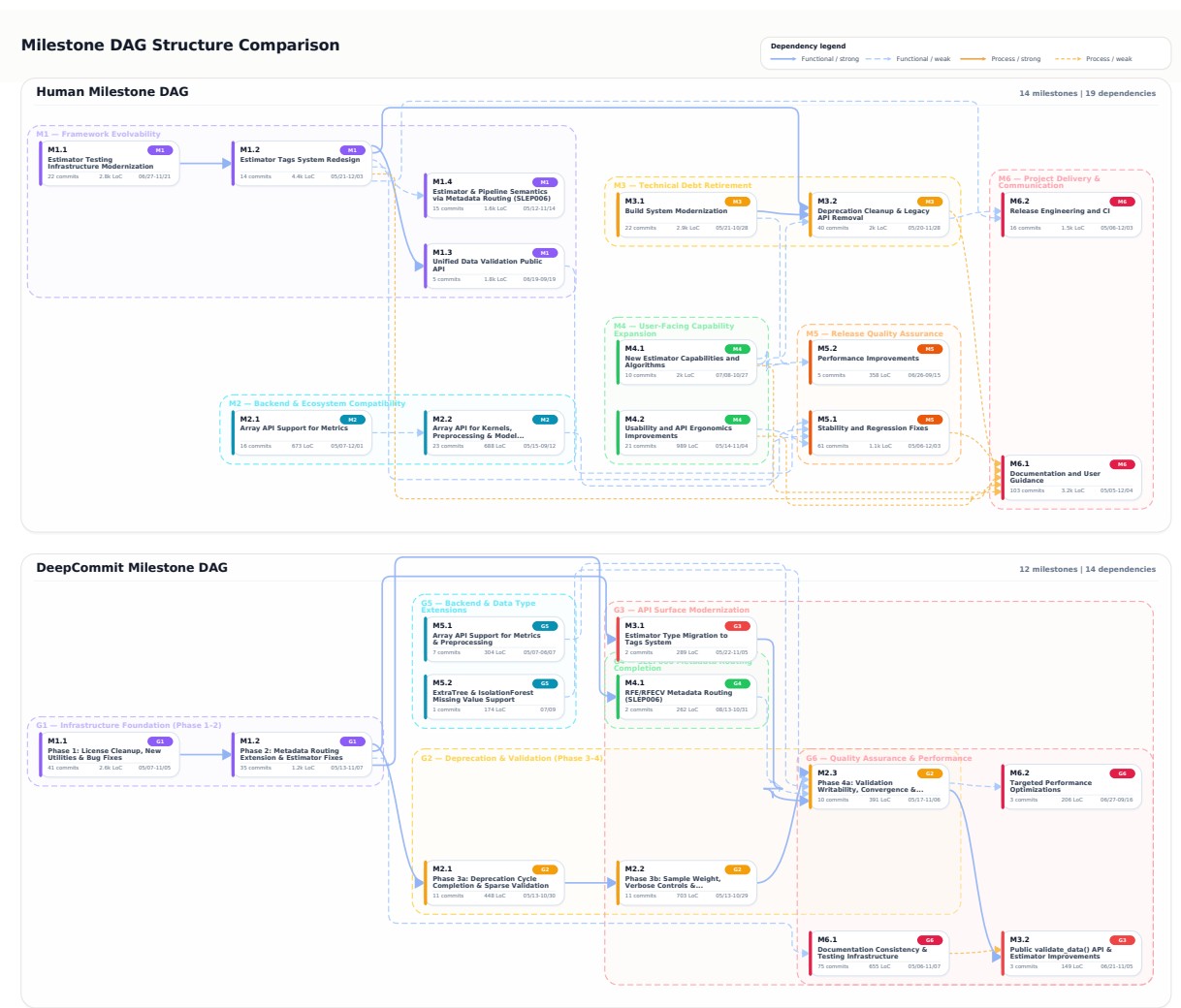

*Figure 28.* Structural comparison of Human and DeepCommit Milestone DAGs. The Human DAG contains 14 milestones and 19 dependencies, while the DeepCommit DAG contains 12 milestones and 14 dependencies. Edges denote functional or process-level dependencies with strong/weak strength.

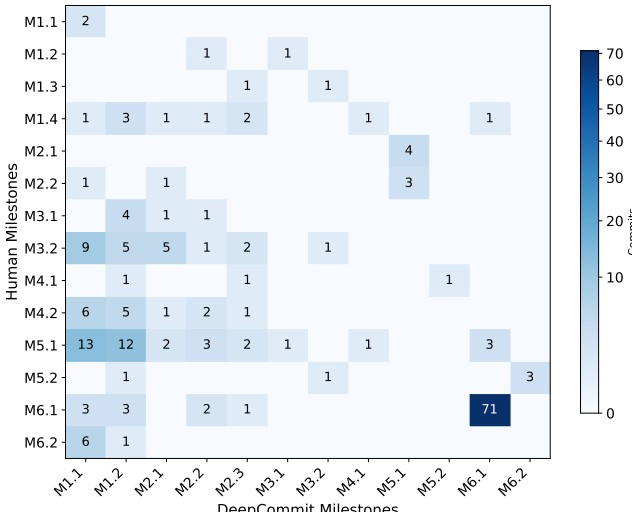

*Figure 29.* Contingency matrix of commit assignments (Human × DeepCommit) for the 201 shared commits. The dominant Human M6.1 ↔ DeepCommit M6.1 block (71 commits) reflects high agreement on documentation, while the dispersed pattern across DeepCommit M1.1–M2.3 reveals how phase-based partitioning fragments intent-based Human milestones.

(14, functional preconditions) from NFR edges (5, process-level dependencies pointing to Documentation) and annotates predominantly weak couplings (17 weak / 2 strong), expressing that most milestones can proceed in parallel. DeepCommit has 8 strong / 6 weak edges, with the main phase chain entirely strong, reflecting strict topological ordering rather than semantic coupling. Similarly, the Human DAG models cross-cutting workflows—five milestones connect to M6.1 via NFR edges ("code first, document later")—while DeepCommit's M6.1 has only 2 incoming edges, since its bottom-up construction does not capture process-level conventions.

### D.7.3. AGREEMENT ANALYSIS AND SUMMARY

Despite moderate overall agreement (ARI = 0.538, Normalized Mutual Information = 0.444), certain regions converge strongly. We measure per-milestone overlap as the fraction of a Human milestone's shared commits that map to a single DeepCommit milestone. Documentation shows the highest overlap: 89% of Human M6.1 commits (71 of 80) land in DeepCommit M6.1. Array API milestones also align well (Human M2.1→DeepCommit M5.1: 100%; Human M2.2→DeepCommit M5.1: 60%), as do performance optimizations (Human M5.2→DeepCommit M6.2: 60%). These high-overlap milestones share distinctive file-modification patterns and isolated module boundaries. Conversely, intent-defined milestones show low overlap—Human M5.1 (Stability Fixes: 35%), M3.2 (Deprecation Cleanup: 39%), M1.4 (Metadata Routing: 30%)—because their commits span multiple modules and time periods, united by purpose rather than code proximity.

In summary, DeepCommit recovers milestone boundaries consistent with human annotation when technical boundaries are clear, but falls back to phase-based partitioning for cross-module, intent-defined work. The Human DAG's structure (strategically coherent groups, FUNC/NFR edge distinction, and cross-cutting workflow modeling) captures developer reasoning that remains difficult to infer from dependency topology alone.

## E. Limitations

SWE-Milestone and the DeepCommit pipeline have several limitations that bound the conclusions to be drawn and the settings to which they currently apply.

**Test-Suite Dependency.** Our construction relies on repositories with well-maintained, executable test suites that provide reliable F2P and P2P signals. Projects that lack rich test coverage, or whose tests depend on inaccessible external services, cannot currently be incorporated into the benchmark.

**Filtering Bias.**  The pipeline retains only commits that touch source code with non-trivial inter-commit dependencies, dropping documentation-only commits and commits without resolvable structural ties. This filtering improves DAG quality and evaluation tractability, but it may bias the resulting benchmark toward dependency-rich evolution and underrepresent independent maintenance work.

**Data Contamination Risk.**  The repositories used in SWE-Milestone are high-impact open-source projects whose commit histories may have appeared in the pretraining corpora of frontier models. The substantial performance gaps we observe among frontier models suggest contamination has limited impact on relative ranking, but we cannot fully rule out memorization on individual milestones. Continuously refreshing the benchmark with newly merged commits, or applying DeepCommit to private repositories, would mitigate this risk.

**Human-in-the-Loop Reliance.**  Two stages of DeepCommit still require human-expert oversight: (i) the *MainAgent*'s scheduling decisions during runtime environment resolution, where humans guide the trade-off between testbed quality and resolution cost, and (ii) the SRS verification stage, where human annotators run the three-step refinement loop described in Appendix C.1. Fully automating these stages remains an open engineering problem.

**Scale Limit.**  The current pipeline targets release ranges whose source-code gold patch is under roughly 30k LoC. Larger ranges produce Milestone DAGs that exceed the agent's resolution budget and frequently fail the testbed-construction gates. Scaling DeepCommit to longer histories will require further improvements to both DAG construction and runtime resolution.

