# OpenReview forum: "EvoClaw: Evaluating AI Agents on Continuous Software Evolution"
_ICML.cc/2026/Conference — ICML 2026 regular_

### Official Review · Reviewer_4d38 · 2026-03-09

**Soundness:** 2
**Presentation:** 2
**Significance:** 3
**Originality:** 3
**Overall Recommendation:** 5
**Confidence:** 4

**Summary:**

The paper proposes a new benchmark for software engineering AI agents, DevEvol. This benchmark positions itself as a middle ground between such existing benchmarks as SWE-bench (operating on singular atomic issues) and Commit-0 (operating on the scale of building the whole application from scratch) by taking a view on the development of a large repository as a series of milestones to evolve through. The authors describe the data gathering procedure and report results for such frontier systems as Claude-Code, Codex, and Gemini-CLI.

**Compliance With Llm Reviewing Policy:**

Affirmed.

**Final Justification:**

In their rebuttal, the authors lifted some of my concerns w.r.t. the presentation. Given this, I consider the paper rather worthy of acceptance.

**Key Questions For Authors:**

- How does the proposed method relate to the ongoing body of research in software evolution, ideally, including that from the software engineering research, and not only machine learning research venues?
- What is the reason for the specific formulation of the _Score_, and how to interpret it?
- How does your proposed repository collection method differ from SWE-EVO?

**Limitations:**

yes

**Strengths And Weaknesses:**

Strengths:
- The paper addresses an important issue in machine learning for software engineering. Especially given the growth of AI agents' autonomy in task execution, which is beyond what is present in SWE-bench, yet below what is expected in Commit-0. Moreover, the results reported in Table 1 seem good in terms of saturation.
- The approach to collecting such type of data seems novel to me.

Weaknesses:
- The paper doesn't describe the procedures in enough detail to estimate how exactly it works and whether it contains any problems. Since there's also no supplementary code, it's very hard to check the proposed methodology for validity. One-sentence descriptions for a whole module that go like "Candidate inter-milestone edges are proposed based on commit-level dependencies, file cochange patterns, and module relationships, and subsequently validated through agent-based semantic reasoning." aren't giving much information on how this was performed. Section 3 reads a lot like a very high-level sketch, rather than as sharing a method with the community.
- The presentation fails to engage with the body of ongoing research in this direction. While the authors rightfully note that the main body of research remains based on small atomic changes and bugfixes, there are middle-ground works at least on two axes.
  - On the axis of change size, there's a lot of middle-ground works like FeatureBench, or SWE-Dev that offer a middle granularity that is close to what the authors propose in Figure 1.
  - On the axis of software evolution, while the authors don't further engage in comparison with SWE-EVO (from the same Figure 1, I infer that this is due to the granularity of change, which the authors deem too coarse), the results reported in SWE-EVO are not so far away (in resolution rate) from what is reported in Table 1. In my opinion, the difference between the proposed benchmarks requires deeper exploration, given their similarity.
- The physical meaning behind the _Score_ introduced by the authors is not discussed.

On a more editorial note:
- The claim about Gemini 3 Pro having fewer interaction turns due to limited RL (around line 314, right column) would benefit from some supporting evidence.
- The beginning of Section 6.2 (first two sentences) is duplicated almost verbatim.
- Making conclusions about the repository itself based on the performance of models on this repository doesn't seem to be correct from the methodological point of view (line 382, left column, and on for next several sentences), from a benchmark paper I'd much rather expect independent evaluation of some properties of the repository, and after that report on how these qualities correlate with model performance.

Overall, the paper is well written, and the idea of measuring the ability of AI agents to perform software evolution via a set of consequent changes is novel and important. However, at the current version, the presentation of the paper doesn't provide the ability to estimate the correctness of the implementation.

---

> ### Author Rebuttal · Authors · 2026-03-30
>
> We thank the reviewer for the constructive feedback and for recognizing the novelty and importance of DevEvol.
>
> ## W1: Methodology Transparency & DAG Construction Validity
>
> In Appendix A/B/C, we already include additional details of our DeepCommit pipeline. **[We share the implementation and prompts for review](https://anonymous.4open.science/r/DevEvol-48A9/DeepCommit/README.md)** and will release them upon acceptance.
>
> DeepCommit is an iterative pipeline validated through three progressively rigorous checks:
>
> - **Structural Validity Check**: We require complete **commit coverage** over the selected range, **preserve commit-level dependencies** at the milestone level, and enforce **DAG acyclicity**; this prevents missing commits and malformed milestone graphs, and ensures basic dependency.
> - **Executability Check**: We reconstruct milestone testbeds by sequentially cherry-picking commits in topological order, and require successful **patch application, build, and test collection** for each environment. If a prerequisite is missing or a commit is misassigned, the issue is exposed here as a patch-apply conflict, compilation failure, or test-collection failure, rather than silently compromising benchmark validity.
> - **Test Behavior Check**: We ensure that the test suite provides milestone-relevant signals. We retain as many tests touched by each milestone’s commits as possible (**87.1%** overall collection), verify the observed START-to-END test transitions are logically consistent with the intended milestone semantics, and filter flaky tests through repeated execution. Every retained milestone with source changes should expose at least one F2P test; otherwise, it is treated as a maintenance task and could be excluded from the final evaluation path.
>
> By enforcing these rigorous checks throughout the pipeline, we ensure that the inevitable noise introduced at intermediate steps does not compromise the final benchmark:
>
> - Segmentation: We successfully restricted the average **milestone LoC variance to 0.96**; thus, any variations mainly affect task difficulty rather than structural soundness.
> - Dependencies: Missing edges are strictly caught by the Executability Check, while redundant edges merely reduce parallelism.
> - Instructions: Risks of underspecification are mitigated via final human verification. Less than **2%** of the remaining impact on the final score (refer to the reply to reviewer wxhr Q1).
>
> Overall, our core philosophy is not to claim a completely noise-free process, but to ensure that the final generated benchmark is structurally sound, executable, and yields reliable evaluation signals. We will incorporate these details into our revised manuscript.
>
> ## W2/Q3 Comparison with Existing Benchmarks
>
> **[This table compares DevEvol with existing benchmarks.](https://anonymous.4open.science/r/DevEvol-48A9/rebuttal_images/benchmark_comparison.png)** The key contribution of DevEvol is not merely its milestone-level granularity, but its explicit modeling of **cross-task dependencies as a Milestone DAG**. While FeatureBench, SWE-Dev, and SWE-EVO all move beyond bug-fixing toward larger development tasks, they still evaluate independent **single-task instances**.
>
> In particular, SWE-EVO strengthens the **long-horizon** aspect of a **single** software-evolution task, whereas DevEvol emphasizes the **continuity of a dependency-constrained sequence** of tasks over a persistent repository history. This makes intermediate progress quantifiable for the first time, allowing DevEvol to directly analyze repository-evolution dynamics and downstream error propagation. Correctly capturing task dependencies requires significant engineering efforts.
>
> ## W3/Q2 Interpret Score and analyze snowball effect
> The original milestone-level $Score$ measures joint success on new-functionality acquisition (F2P) and regression preservation (P2P), but as a single aggregate number, it does not by itself explain the source of streaming degradation. We therefore additionally report a Recall/Precision-based decomposition. For milestone $m$,
>
> $$
> \mathrm{Recall}\_m=\frac{N\_{\mathrm{fixed},m}}{N\_{\mathrm{required},m}},
> \qquad
> \mathrm{Precision}\_m=\frac{N\_{\mathrm{fixed},m}+\epsilon}{N\_{\mathrm{fixed},m}+N\_{\mathrm{broken},m}+\epsilon}.
> $$
>
> Recall measures how much required functionality has been acquired, while Precision measures how much previously acquired functionality is preserved. **[In the new Fig. 9–10](https://anonymous.4open.science/r/DevEvol-48A9/rebuttal_images/recall-precision_error-chains.png)**, this decomposition shows that **Recall continues to grow approximately linearly, whereas Precision saturates much earlier**, and unresolved regressions repeatedly propagate into downstream milestones. Together, these results indicate that the streaming gap is driven primarily by regression accumulation, rather than simply by degraded implementation of later milestones.

---

> > ### Author Rebuttal · Reviewer_4d38 · 2026-04-02
> >
> > I thank the authors for the constructive rebuttal.
> >
> > Overall, I consider my questions to be rather addressed (the additional metric analysis is particularly appreciated). I will increase the score, but ask the authors to describe their methodology and comparison to other benchmarks more specifically in the main body of the paper.

---

> > > ### Author Response · Authors · 2026-04-02
> > >
> > > Thank you so much! We promise to carefully refine the method section and add more comparisons with existing benchmarks based on each reviewer's suggestions. Also, with a more comprehensive and in-depth experimental analysis.

---

### Official Review · Reviewer_S1nu · 2026-03-10

**Soundness:** 3
**Presentation:** 3
**Significance:** 4
**Originality:** 4
**Overall Recommendation:** 5
**Confidence:** 5

**Summary:**

This paper points out that current evaluations of Large Language Model (LLM) coding agents predominantly focus on static, independent tasks, which fails to reflect the continuous and sequentially dependent nature of real-world software evolution . To address this gap, the authors propose two core contributions: first, DeepCommit, an automated agentic pipeline capable of extracting structured and verifiable "Milestone Directed Acyclic Graphs (DAGs)" from noisy Git histories; second, DevEvol, a streaming evaluation benchmark based on these DAG data, which requires agents to process streaming tasks and maintain codebase states within a continuous development flow . By comparing independent task evaluation with streaming evaluation, the paper reveals that current frontier models suffer severe performance degradation due to the downstream propagation of early errors (the "snowball effect") when facing long-horizon contexts and technical debt accumulation .

**Compliance With Llm Reviewing Policy:**

Affirmed.

**Key Questions For Authors:**

1. Regarding Benchmark Scalability and Data Contamination: Currently, DevEvol contains 96 milestones across 7 repositories and relies heavily on human experts for final SRS verification . Can the authors discuss how the DeepCommit workflow could be further scaled up to support the generation of large-scale Reinforcement Learning (RL) training data for agents? Furthermore, given that the selected targets are high-impact open-source repositories, it is highly likely that the evaluated closed-source frontier models have already "seen" these real code commit histories during their pre-training phase. Under this dynamic evaluation framework, do the authors have mechanisms (or future extension plans) to measure and prevent this potential Data Contamination from artificially inflating the evaluation results?

2. Regarding Specific Mechanisms for Interface Alignment: The paper mentions generating the SRS from gold patches via reverse engineering and strictly aligning it with F2P tests . In automated evaluation, if an Agent implements completely correct functional logic but uses a different interface naming convention than expected, causing the test script call to fail, how does the system judge this failure caused by "contract misalignment" rather than a "logical error"? Could you provide a specific example of how this is circumvented?

3. Regarding the Impact of Filtered Commits: In the Case Study of Section 6.4, DeepCommit filtered out approximately 46% of the commits (primarily low inter-dependency activities) . Does this strict pursuit of topological structure artificially simplify the benchmark? In real-world development, agents are equally required to handle these fragmented tasks that are interspersed among core functionalities.

**Limitations:**

No.
The authors provide an Impact Statement at the end of the main text discussing macro-level societal impacts , and they deeply explore the structural deviations between the automated DAG and the human-annotated DAG in the Case Study section . However, the paper lacks a concise, formal summary specifically addressing its core technical limitations. It is recommended that the authors add a brief discussion explicitly noting the method's strong reliance on the target repository having an existing high-quality test suite, the benchmark bias introduced by filtering commits with no obvious dependencies , and the unavoidable risk of Data Contamination when evaluating well-known open-source repositories. This does not need to be lengthy, but explicitly stating these boundaries will further elevate the rigor of the paper.

**Strengths And Weaknesses:**

Soundness
* Strengths: The methodology for data construction is highly solid. The DeepCommit pipeline effectively combines low-level static analysis with LLM semantic reasoning to filter out noise from Git histories . Furthermore, the authors implement strict quality control during the test collection phase, including removing flaky tests and distinguishing between Fail-to-Pass (F2P) and Pass-to-Pass (P2P) tests , which makes the final evaluation metrics highly credible.
* Weaknesses: Employing reverse engineering to generate Software Requirement Specifications (SRS) is logically self-consistent; however, when dealing with highly complex interface refactoring, fully automated environment construction may still suffer from alignment deviations. Additionally, as the authors candidly point out in the Case Study, the DAG constructed based on topological dependencies filters out a large number of low-dependency commits, which deviates somewhat from human development intuition based on "semantic intent". Lastly, because Human-in-the-loop verification is required in the final stage to guarantee task solvability, the scaling of this pipeline faces an implicit human-labor bottleneck.

Presentation
* Strengths: The motivation of the paper is exceptionally clear, directly addressing the pain points of current static snapshot benchmarks. The chart designs (especially the granularity comparison in Figure 1 and the cumulative error analysis in Figure 6) intuitively and effectively support the core arguments.
* Weaknesses: The primary cognitive load of this paper stems from its extremely dense, continuous abstract descriptions. The entire workflow (from raw commit filtering $\rightarrow$ building the underlying dependency graph $\rightarrow$ consolidating into a milestone DAG $\rightarrow$ reconstructing the executable environment based on topological order) forms a very long and abstract conversion chain . If a reader fails to fully grasp a specific level of abstraction (e.g., node consolidation or dependency inference), it becomes very difficult to understand the operational logic of the subsequent streaming evaluation framework. It is strongly recommended that the authors introduce a concrete, end-to-end "Running Example" when introducing the DeepCommit pipeline to connect these abstract stages.

Significance
* Strengths: This work is highly inspiring and pushes the field forward. It shifts the evaluation of coding agents from "isolated bug fixing" to "long-horizon codebase maintenance," a scenario much closer to real-world industrial needs . The revealed "snowball effect" and the low real-world resolve rate (~10%) illuminate clear paths for future research regarding the deficiencies of Agentic AI in state management and long-term planning capabilities.

Originality
* Strengths: This paper demonstrates strong originality in its evaluation paradigm. Introducing a DAG-based "streaming task unlocking" mechanism and requiring agents to work continuously in a codebase with persistent states is highly innovative among existing code generation benchmarks . Reconstructing executable historical environments via topological sorting is also a very clever data engineering innovation .

---

> ### Author Rebuttal · Authors · 2026-03-31
>
> We sincerely thank the reviewer for raising these important questions and suggestions. In the revision, we will add a more explicit discussion of the limitations together with concrete examples to better explain DeepCommit. We also appreciate that the reviewer has highlighted several key issues that should be discussed more clearly in the paper, and we address them below.
>
> ## Scalability for RL Training Data
>
> Scaling DeepCommit from a quality-prioritized evaluation benchmark to a large-scale RL data generator primarily requires balancing verification strictness with automation and throughput. We see three practical directions to achieve this.
>
> - **Fully Automated SRS Verification:** The human-involved SRS verification is already heavily agent-assisted and can be further automated. For static analysis, two agents can iteratively debate whether the SRS is overly detailed (leaking implementation) or underspecified. For dynamic analysis, multiple task agents can dry-run the SRS, while an evaluator agent decides whether to revise the SRS based on failed tests. This suggests that DeepCommit has the potential to remove humans from the loop while maintaining reasonably high SRS quality.
> - **Relaxing DAG Quality Control:** When scaling up, the current pipeline can run into repeated timeouts and refinement loops because of limited agent capability and strict quality-control requirements. One practical solution is to relax some quality-control requirements and reduce DAG complexity, for example, by processing smaller commit windows at a time or filtering out more commits.
> - **Reducing Iterative Cost:** The current pipeline remains expensive because it requires multiple rounds of refinement. We can reduce this cost by using smaller, specialized models for parts of the verification pipeline, or by using stronger models to reduce the number of refinement rounds.
>
> ## Preventing Data Contamination
>
> We acknowledge that frontier models may have been trained on these high-impact open-source repositories. However, DeepCommit restructures raw commit histories into Milestone DAGs and reverse-engineers them into SRS-driven tasks. Solving these tasks requires not only memorization but long-horizon codebase development and maintenance. The substantial performance gaps among frontier models suggest that contamination has a limited impact on model evaluation.
>
> We plan to prevent contamination through the following ways.
>
> - **Continuous Live Updates**: We can continually refresh the benchmark with newly merged commits from recent development windows, creating tasks that are necessarily unseen by current models. However, this may introduce some variance in difficulty and dependencies due to the limited window size.
> - **Hidden Test Sets via Private Repositories:** The best solution is to deploy DeepCommit on private production repositories. This would enable a fully hidden evaluation set with complete repository evolution histories.
>
> ## Interface alignment
>
> As detailed in our reply to Reviewer wxhr (W1), we use a strict multi-stage SRS verification process to minimize SRS–Test misalignment and include necessary interface-level details whenever needed for solvability.
>
> During automated evaluation, the overall score calculation does not distinguish between failures caused by "contract misalignment" and "logical errors". This is because, as long as these interface names can be reasonably inferred from the SRS or codebase's conventions, the agent is expected to get them right to ensure compatibility with the overall ecosystem.
>
> Although our automated metric does not distinguish between these errors, we conducted a post hoc log analysis of Opus 4.6 to quantify the impact of "contrat misalignment." We found that fewer than 1.2% of test errors are caused by contract misalignment, suggesting that interface ambiguity in the verified SRS is rare and that contract misalignment has minimal impact on the overall score.
>
> For example, the expected file name `useLibrarySelection.js` was already inferable from tests in the base environment. The agent implemented the correct hook logic but placed it in `useLibraryState.js`, causing import-resolution failures that a simple rename would fix. This is therefore a contract misalignment rather than a logical error.
>
> ## Impact of filtered Commits
>
> Our filtering is primarily a validity decision, not a simplification trick: DevEvol focuses on source-code evolution with reliable executable signals, while many non-source commits (e.g., modify static assets, data files, demos) contribute weakly verifiable changes and substantial incidental patch dependencies. Retaining them would often linearize the milestone DAG along commit time rather than reveal meaningful functional structure. We further remove milestones with insufficient F2P tests, trivial difficulty, or weak dependency relevance. Thus, filtering mainly improves quality and keeps DAG construction tractable, rather than artificially simplifying the benchmark.

---

> > ### Author Rebuttal · Reviewer_S1nu · 2026-04-02
> >
> > Thank you to the authors for the detailed and constructive rebuttal. The proposed directions for scalability and mitigating data contamination (e.g., continuous live updates and private repositories) are well-reasoned. I particularly appreciate the post-hoc log analysis showing that contract misalignment accounts for < 1.2% of test errors, which completely resolves my concern regarding interface alignment. The rationale behind filtering commits to ensure valid executable signals is also clear.
> >
> > I will maintain my positive assessment of this paper and keep my score.

---

> > > ### Author Response · Authors · 2026-04-02
> > >
> > > Thank you so much for the valuable feedback! We will further refine our work based on your concerns and suggestions.

---

### Official Review · Reviewer_wxhr · 2026-03-10

**Soundness:** 3
**Presentation:** 2
**Significance:** 3
**Originality:** 3
**Overall Recommendation:** 4
**Confidence:** 3

**Summary:**

This paper introduces DevEvol, a benchmark for evaluating LLM agents on continuous, dependency-driven software evolution rather than isolated snapshot tasks. The authors construct milestone DAGs from real-world repositories using the DeepCommit pipeline and evaluate agents under both independent and streaming settings.
Results show a substantial performance gap between isolated milestone evaluation and streaming evaluation, which the authors attribute to error accumulation over long development sequences. The work aims to highlight limitations of snapshot-based benchmarks and advocate for more realistic long-horizon evaluation.

**Compliance With Llm Reviewing Policy:**

Affirmed.

**Final Justification:**

I believe the current paper has reached the bar for weak acceptance, although there are some minor issues.

**Key Questions For Authors:**

### Key Questions for Authors

1. **Realism and granularity of reconstructed milestone instructions.**
DevEvol reconstructs milestone-level task descriptions from historical commits rather than using naturally occurring issue reports. Could the authors provide additional analysis on the generated SRS instructions, such as their level of abstraction, whether they explicitly reference implementation details (e.g., function or file names), and how closely they resemble real development objectives? A clearer characterization of instruction granularity would help determine whether the benchmark difficulty reflects long-horizon reasoning challenges rather than artifacts of instruction design.

2. **Validation and noise analysis of milestone DAG construction.**
The paper mentions three layers of validation for milestone construction and dependency extraction. Could the authors provide more quantitative evidence on the reliability of this process—for example, estimates of error rates in milestone segmentation, dependency extraction accuracy, or manual validation statistics? Since the benchmark is synthesized from reverse-engineered requirements and automatically constructed DAGs, additional transparency on validation quality would strengthen confidence in the benchmark.

3. **Prompting protocol and preservation requirements in the streaming setting.**
In the streaming evaluation, agents are evaluated on both implementing new functionality (F2P tests) and preserving previously satisfied functionality (P2P tests). Could the authors clarify whether milestone prompts explicitly instruct the agent to maintain all previously implemented functionality? If regression avoidance is not explicitly specified, to what extent might the observed streaming degradation reflect a mismatch between the optimization objective implied by the prompt and the evaluation criteria?

4. **Analysis of the claimed “snowball effect.”**
The paper attributes the performance gap between independent and streaming settings to a “snowball effect” of accumulated errors. Could the authors provide a breakdown of failures across F2P (new functionality) and P2P (regression) tests over time? Such an analysis would help clarify whether the degradation primarily arises from regression accumulation, difficulty implementing later milestones, or other factors, and would strengthen the causal interpretation of the results.

**Limitations:**

yes

**Strengths And Weaknesses:**

### Strengths

- **Well-motivated problem setting.**
  The paper addresses a meaningful gap in current software engineering benchmarks by moving beyond isolated snapshot tasks toward continuous, dependency-aware evaluation. The motivation is timely and grounded in realistic development workflows.

- **Challenging and practically relevant task formulation.**
  The proposed streaming setting captures aspects of long-horizon code evolution that are underexplored in existing benchmarks, and the observed performance gap highlights a non-trivial challenge for current LLM agents.

- **Substantial engineering effort.**
  The DeepCommit pipeline, milestone DAG construction, testbed generation, and multi-stage validation indicate considerable implementation effort. The overall methodology appears thoughtfully designed.

- **Potential to influence future evaluation protocols.**
  If the benchmark construction and evaluation details are further clarified and strengthened, this framework could serve as a valuable step toward more realistic long-horizon agent evaluation.

### Weaknesses

1. **Potential realism issues in reverse-engineered task instructions.**
Unlike SWE-bench, where tasks are grounded in naturally occurring issue reports written before implementation, DevEvol reverse-engineers milestone-level requirements from historical commits. This raises concerns regarding the realism and consistency of the generated task instructions.
In particular, the difficulty of agent evaluation is highly sensitive to instruction granularity. If the reconstructed SRS descriptions enumerate concrete implementation details (e.g., explicitly naming functions, files, or specific code-level changes), the task may reduce to localized patch reconstruction. Conversely, if the descriptions are abstract or underspecified, agents may fail due to ambiguity rather than dependency accumulation.
The paper does not clearly quantify or analyze this trade-off, nor does it provide evidence that the generated instructions faithfully resemble real-world development objectives. As a result, it is unclear whether performance differences—especially in the streaming setting—reflect genuine long-horizon reasoning challenges or artifacts of instruction formulation.

2. **Limited transparency in benchmark validation.**
The paper states that milestone construction and dependency extraction undergo three layers of validation. However, given that DevEvol is a synthesized benchmark built from reverse-engineered requirements and automatically constructed DAGs, validation constitutes a central component of its credibility rather than a peripheral detail.
The description of these validation stages is relatively high-level, and the paper does not quantify potential noise in milestone segmentation, dependency extraction, or instruction reconstruction.

3. **Possible misalignment between task prompts and evaluation criteria.**
In the streaming setting, agents are evaluated not only on implementing new functionality (F2P tests) but also on preserving previously satisfied functionality (P2P tests). However, it is unclear whether milestone instructions explicitly emphasize the requirement to maintain all prior functionality.
If preservation constraints are not clearly specified in the task prompt, agents may optimize for the immediate milestone objective without considering regression risks. In such cases, performance degradation in the streaming setting may partially reflect objective–evaluation misalignment rather than genuine long-horizon reasoning difficulty.
Clarifying the exact prompting protocol and whether regression avoidance is explicitly required would strengthen the validity of the streaming evaluation.

4. **Limited analysis supporting the “snowball effect.”**
The central claim of the paper is that the streaming setting reveals a “snowball effect,” where errors accumulate over time. However, the reported results primarily present aggregate success rates.
Since milestone evaluation consists of both F2P (new functionality) and P2P (regression preservation) tests, separating these two components is crucial to understanding the source of performance degradation. Without disaggregated statistics, it is unclear whether the observed gap between independent and streaming evaluation arises from regression accumulation (P2P failures), degraded new feature implementation (F2P failures), or other confounding factors.
A more detailed breakdown would significantly strengthen the causal interpretation of the streaming degradation claim.

5. **Limited repository diversity.**
DevEvol is constructed from seven repositories, yielding 96 milestones organized into evolution DAGs. While the milestone count is non-trivial, the number of independent evolutionary trajectories remains small. Since milestones within a repository are structurally dependent, the effective diversity of development contexts is limited. This raises questions about the generalizability of the reported streaming degradation phenomenon across broader software domains.

6. **Minor formatting issues.**
The manuscript appears to contain minor formatting inconsistencies. For example, the first page includes a footnote marker without content, and page 8 contains noticeable unused space. These issues do not affect the technical content but may require minor formatting adjustments in the final version.

---

> ### Author Rebuttal · Authors · 2026-03-28
>
> We appreciate the reviewer for recognizing DevEvol's potential impact and constructive feedback.
>
> ## W1/Q1: Realism and Verification of SRS
>
> ### Realism
>
> Our SRS is designed as a **minimally solvable** behavior/contract-level specification, not a patch-reconstruction hint.
> - **It avoids line-level edits and patch-like instructions**, and files/functions names unless such identifiers are themselves part of the public contract or necessary to make the SRS unambiguous.
> - **It includes enough information to solve the task**. Each milestone’s SRS contains several Feature Requirements, each specifying:
>     - Problem: symptoms and context
>     - Requirements: the intended functionality and constraints in an implementation-agnostic way
>     - Acceptance: observable pass criteria without leaking test details
>
> Compared with raw issue reports, our SRS is consistent with emerging specification-driven development workflows for coding agents. See an example in **[zeromicro/go-zero M001 SRS](https://anonymous.4open.science/r/DevEvol-48A9/DevEvol_data/zeromicro_go-zero_v1.6.0_v1.9.3/srs/M001/SRS.md)**.
>
> ### Verification
>
> After agents reverse-engineer the SRS, human verification refines it by enforcing three deliberately competing principles:
>
> - **Specify what is required, not how to implement it**: state motivation, required behavior, and acceptance criteria without prescribing the implementation.
> - **Align with test intent, not test artifacts**: cover the evaluated behaviors without revealing test names, concrete inputs, or exact asserted values, while allowing only necessary public interface constraints.
> - **Ensure solvability**: include the essential formats, function signatures, edge cases, and non-inferable conventions needed to derive a correct implementation.
>
> We further perform **two rounds** of human-in-the-loop verification. First, human annotators use a well-defined questionnaire, together with agent-assisted workflows, to identify and fix SRS issues through the following loop:
>
> 1. **Static verification**: iteratively check whether the SRS satisfies the principles above until no further obvious misalignment remains.
> 2. **Independent-task dry-run verification**: iteratively evaluate each milestone as an independent task using strong coding agents with the refined SRS. After each run, annotators inspect failed tests and revise the SRS only by adding the minimal non-inferable conventions required for solvability. After several iterations, state-of-the-art agents typically achieve roughly **80–90% scores** on independent tasks, which supports that the SRS is solvable rather than substantially under-specified.
> 3. **Continuous-task dry-run verification**: iteratively evaluate DAG-level continuous tasks to surface missing cross-milestone dependencies and residual SRS gaps that only emerge under continuous execution.
>
> After this process, we conduct a **second end-to-end review by an independent senior expert** to further validate the refined SRS.
>
> In practice, the refinement process converges as most SRS errors can be localized and corrected through repeated evaluation. The first refinement round can change scores by around **5%**, whereas after the second round the impact typically falls within **2%**, suggesting that residual noise has a limited effect on overall ranking.
>
> After verification, we refined 71% of milestones, mainly by removing implementation leakage (39%), removing test leakage (31%), refining inaccurate or underspecified specifications (25%), and adding missing functional requirements (5%).
>
> ## Q2/W2: Validation and noise analysis of milestone DAG construction
> Please refer to the reply to the reviewer 4d38 W1.
>
> ## Q3/W3: Prompting protocol and preservation requirements
>
> **[Our prompt](https://anonymous.4open.science/r/DevEvol-48A9/DevEvol/harness/e2e/prompt/v2.md)** explicitly instructs the agent to maintain consistency across prior changes and preserve previously implemented functionality. **[This is further supported by our Fig. 9 Precision analysis](https://anonymous.4open.science/r/DevEvol-48A9/rebuttal_images/recall-precision_error-chains.png)**: stronger models achieve higher Precision (P2P maintenance), suggesting that the prompt is not ambiguous.
>
> ## Q4/W4: Analysis of "snowball effect"
> Please refer to the reply to the reviewer 4d38 W3/Q2.
>
> ## W5 Limited repository diversity
> DevEvol is highly generalizable despite relying on 7 repositories, as it spans 5 programming languages, averaging ~6k LoC per repo,  and is fundamentally driven by a universal LLM bottleneck — regression accumulation. Because long-horizon tasks amplify performance gaps, these 7 human-validated DAGs already stably separate frontier models while strictly maintaining an accessible per-run cost of ~$500 with Opus 4.5 (comparable to SWE-bench Verify).

---

> > ### Author Rebuttal · Reviewer_wxhr · 2026-04-02
> >
> > I believe the author's current rebuttal has largely addressed my concerns. Please address the writing and formatting issues in the future.

---

> > > ### Author Response · Authors · 2026-04-02
> > >
> > > Thank you so much! We apologize for the inconvenience and promise to carefully fix all the writing and formatting issues in our next version.

---

### Decision · Program_Chairs · 2026-04-30

**Decision:**

Accept (regular)

**Comment:**

The paper introduces a benchmark (DevEvol) and a method (DeepCommit pipeline) to evaluate LLM coding agents in a streaming, long-horizon software evolution setting. It finds a large gap between performance of coding agents on snapshots vs continuously evolving code repositories.
The problem is very well motivated, and the proposals are novel and significant. The reviewers all agreed that the evaluation is technically sound, but had questions about the exposition and concerns about methodology transparency. The authors convincingly addressed all the concerns, provided a Precision/Recall decomposition that attributes the observed gap between snapshot and streaming performance to regression accumulation, and committed to releasing implementations for reproducibility. The work contributes a valuable, potentially high-impact benchmark that studies a hitherto under-studied long-horizon failure mode in coding agents.